# An antibody against L1 cell adhesion molecule inhibits cardiotoxicity by regulating persistent DNA damage

Jae-Kyung Nam[1,2,10], A-Ram Kim[1,10], Seo-Hyun Choi[1,3,10], Ji-Hee Kim[1,2], Kyu Jin Choi[1], Seulki Cho[4], Jae Won Lee [5], Hyun-Jai Cho[5], Yoo-Wook Kwon [6], Jaeho Cho[7], Kwang Seok Kim [1], Joon Kim [2], Hae-June Lee[1], Tae Sup Lee[8], Sangwoo Bae[1], Hyo Jeong Hong[4,9✉] & Yoon-Jin Lee [1✉]

Targeting the molecular pathways underlying the cardiotoxicity associated with thoracic irradiation and doxorubicin (Dox) could reduce the morbidity and mortality associated with these anticancer treatments. Here, we find that vascular endothelial cells (ECs) with persistent DNA damage induced by irradiation and Dox treatment exhibit a fibrotic phenotype (endothelial–mesenchymal transition, EndMT) correlating with the colocalization of L1CAM and persistent DNA damage foci. We demonstrate that treatment with the anti-L1CAM antibody Ab417 decreases L1CAM overexpression and nuclear translocation and persistent DNA damage foci. We show that in whole-heart–irradiated mice, EC-specific p53 deletion increases vascular fibrosis and the colocalization of L1CAM and DNA damage foci, while Ab417 attenuates these effects. We also demonstrate that Ab417 prevents cardiac dysfunction-related decrease in fractional shortening and prolongs survival after whole-heart irradiation or Dox treatment. We show that cardiomyopathy patient-derived cardiovascular ECs with persistent DNA damage show upregulated L1CAM and EndMT, indicating clinical applicability of Ab417. We conclude that controlling vascular DNA damage by inhibiting nuclear L1CAM translocation might effectively prevent anticancer therapy-associated cardiotoxicity.

[1] Division of Radiation Biomedical Research, Korea Institute of Radiological & Medical Sciences, Seoul, Korea. [2] Laboratory of Biochemistry, Division of Life Sciences, Korea University, Seoul, Korea. [3] Department of Surgery, Memorial Sloan Kettering Cancer Center, NY, USA. [4] Division of Biomedical Convergence, College of Biomedical Science, Kangwon National University, Chuncheon, Korea. [5] Biomedical Research Institute, Seoul National University Hospital, Seoul, Korea. [6] Cardiovascular Center & Department of Internal Medicine, Seoul National University Hospital, Seoul, Korea. [7] Department of Radiation Oncology, Yonsei University College of Medicine, Seoul, Korea. [8] Division of RI Convergence Research, Korea Institute of Radiological & Medical Sciences, Seoul, Korea. [9] Scripps Korea Antibody Institute, Chuncheon, Korea. [10] These authors contributed equally: Jae-Kyung Nam, A-Ram Kim, Seo-Hyun Choi. ✉email: hjhong@kangwon.ac.kr; yjlee8@kirams.re.kr

Radiation therapy and the anthracycline chemotherapy drug doxorubicin (Dox; trade name: Adriamycin) are commonly prescribed anticancer treatments[1]. However, the DNA damage caused by these therapies is associated with cardiotoxicity, which causes congestive heart failure and cardiovascular complications and limits the use of these therapies[1]. Indeed, a meta-analysis revealed excess mortality owing to heart disease in women receiving radiation therapy for left-sided breast cancer[2,3]. In the case of Dox, the risk and severity of cardiotoxicity is dose dependent[4,5] and dosage increases to improve cancer outcomes are often associated with increased cardiovascular morbidity and mortality[6,7]. Targeting the molecular pathways underlying this cardiotoxicity could help prevent or control such cardiac complications after anticancer therapy. As a chelator of intracellular iron, dexrazoxane, an FDA-approved drug, prevents Dox-induced heart failure from interfering with the anti-tumour effect of Dox[8]. A recent study also reported that a small molecule allosteric inhibitor of BAX and a compound that stabilises tetrameric PKM2 prevent Dox-induced cardiomyopathy[9,10]; However, there are still no clinical therapies to efficiently prevent Dox-induced cardiotoxicity.

After radiation therapy, damage to the cardiac microvasculature occurs, potentially within days or months[11]. Radiation-induced microvascular ischaemia disrupts capillary endothelial cells (ECs), and injury to differentiated cardiomyocytes results in collagen deposition and fibrosis, which progress to cardiovascular disease[12]. In addition, endothelial dysfunction, inflammation, DNA damage, oxidative stress, and altered coagulation and platelet activity may play important roles in radiation-induced cardiovascular effects[13,14]. The pathogenesis of Dox cardiotoxicity has been suggested to involve DNA damage, transcriptome alteration, mitochondrial iron accumulation, mitochondrial damage, vascular endothelial injury, and reactive oxygen species (ROS) accumulation[15,16].

L1 cell adhesion molecule (L1CAM; also known as CD171) is a transmembrane glycoprotein belonging to the Ig superfamily of cell adhesion molecules (CAMs). It consists of a cytoplasmic intracellular domain, a transmembrane domain, and an extracellular region comprising six Ig-like domains and five fibronectin type III domains[17,18]. The L1CAM extracellular domains interact with other L1CAM molecules, growth factor receptors, integrins, neuropilin-1, and extracellular matrix proteins[19,20]. L1CAM has been reported to exhibit important functions in different cells and tissues, including neurites, kidney tubule epithelial cells, lymphoid and myelomonocytic cells, and intestinal crypt cells[17,21–25]. NCAM plays an important role in cell adhesion through heterophilic interactions with L1CAM. Moreover, neural cell adhesion molecule (NCAM) was found to be upregulated upon metabolic stress in cardiomyocytes[26] and reported to be associated with cardiac vascular development[27]. NCAM genetic variants have been demonstrated to be important heritable predictors of cardiovascular disease, such as hypertension[28]. During hypertension-induced LV remodelling, which leads to heart failure, NCAM was found to be highly expressed[27]. Notably, Nagao et al.[29] showed that NCAM may be involved in LV remodelling with heart failure in a study conducted using 64 human cardiac tissue samples of patients with dilated cardiomyopathy. Further, L1CAM has been shown to have a function in cancer, as it promotes the proliferation, migration, invasion, and chemoresistance of tumour cells[30,31], and its expression is correlated with poor prognosis and metastases in various cancers[19]. Moreover, it has been reported that the L1CAM intracellular domain is cleaved from the membrane-bound portion of L1CAM by ADAM10 and γ-secretase, after which it is translocated to the nucleus to regulate the expression of genes involved in tumour progression[32]. Additionally, nuclear L1CAM signalling augments the DNA damage checkpoint response and radio-resistance of glioblastoma stem cells[33]. It has been reported that L1CAM regulates tumour vascular permeability and endothelial–mesenchymal transition (EndMT) in pancreatic carcinoma[34]. To study anti-L1CAM antibodies as anticancer agents, we previously developed a human mAb, Ab417, that binds to human and mouse L1CAM with high affinity[35,36]. Here, we show the involvement of L1CAM in the DNA damage response in vascular ECs and the therapeutic efficacy of Ab417 against DNA damage-induced cardiotoxicity.

## Results

**Nuclear L1CAM is involved in persistent DNA damage in ECs.** We have previously reported that vascular ECs undergo EndMT during radiation-induced atherosclerosis and pulmonary fibrosis[37,38]. Additionally, CAMs on ECs have been reported to play an important role in radiation-induced vascular damage and EndMT[39,40]. To explore the roles of CAMs in regulating radiation-induced cardiovascular disease, we performed microarray analysis on cardiac ECs. We found that *L1CAM* expression in cardiac ECs was upregulated by 2.1-fold 12 h after irradiation (Supplementary Fig. 1a). In addition, irradiation increased the protein expression of L1CAM in vascular ECs along with that of α-SMA (a fibroblastic marker) and VCAM-1 (an important CAM involved in leucocyte trafficking) (Supplementary Fig. 1b), indicating that EndMT is occurring in response to irradiation. Importantly, these effects were efficiently inhibited when L1CAM was knocked down with small-interfering RNA ((siRNA); Supplementary Fig. 1b), suggesting that L1CAM is involved in EndMT in response to irradiation. To further confirm these observations, we next examined the effects L1CAM inhibition on irradiation-induced transendothelial migration of human monocytes, a marker of the inflammatory response. We found that treatment with the anti-L1CAM mAb Ab417 significantly inhibited the transendothelial migration 24 h after irradiation (Supplementary Fig. 1c). These data indicate that L1CAM may be an effective regulator of radiation-induced vascular damage.

Next, we examined whether radiation-induced DNA damage can affect EndMT. Immunofluorescence experiments in human umbilical vein endothelial cells (HUVECs) indicated that the severity of the elongated fibrotic phenotype (marked by phalloidin, showing actin cytoskeleton reorganisation was correlated with the number of DNA damage foci (marked by γ-H2AX) 48 h after 2 Gy, 5 Gy or 10 Gy of irradiation, and this correlation was dose dependent (Fig. 1a). This was accompanied by dose-dependent colocalization of nuclear L1CAM-C-terminus (L1-CT) with γ-H2AX foci, a marker of DNA double-strand breaks (Fig. 1b). Furthermore, RNA-sequencing (RNA-seq) analysis showed that the expression of EndMT-related genes was higher in 10 Gy-irradiated cells 72 h after irradiation than in 10 Gy-irradiated cells 5 h after irradiation or in non-irradiated cells (Fig. 1c). The average number of γ-H2AX foci per cell at least 70 cells per field increased from 1 to 59 at 1 h post-irradiation, and then decreased to 26 at 48 h post-irradiation (Fig. 1d), indicating persistent DNA damage in the cells. However, the intensity and foci diameter were significantly decreased at 24 h and after increased again thereafter (Supplementary Fig. 2a). The number of γ-H2AX foci with an intensity larger than 40 and a foci diameter of 0.1 μm was counted (Fig. 1d). Live-cell imaging was used to confirm the persistence of DDR foci, and 53BP1 DNA damage foci were traced in HUVECs transfected with GFP-53BP1 (a marker of DNA damage foci); Kaplan–Meier survival curves for GFP-positive foci were analysed from 12 h after 10 Gy irradiation, showing

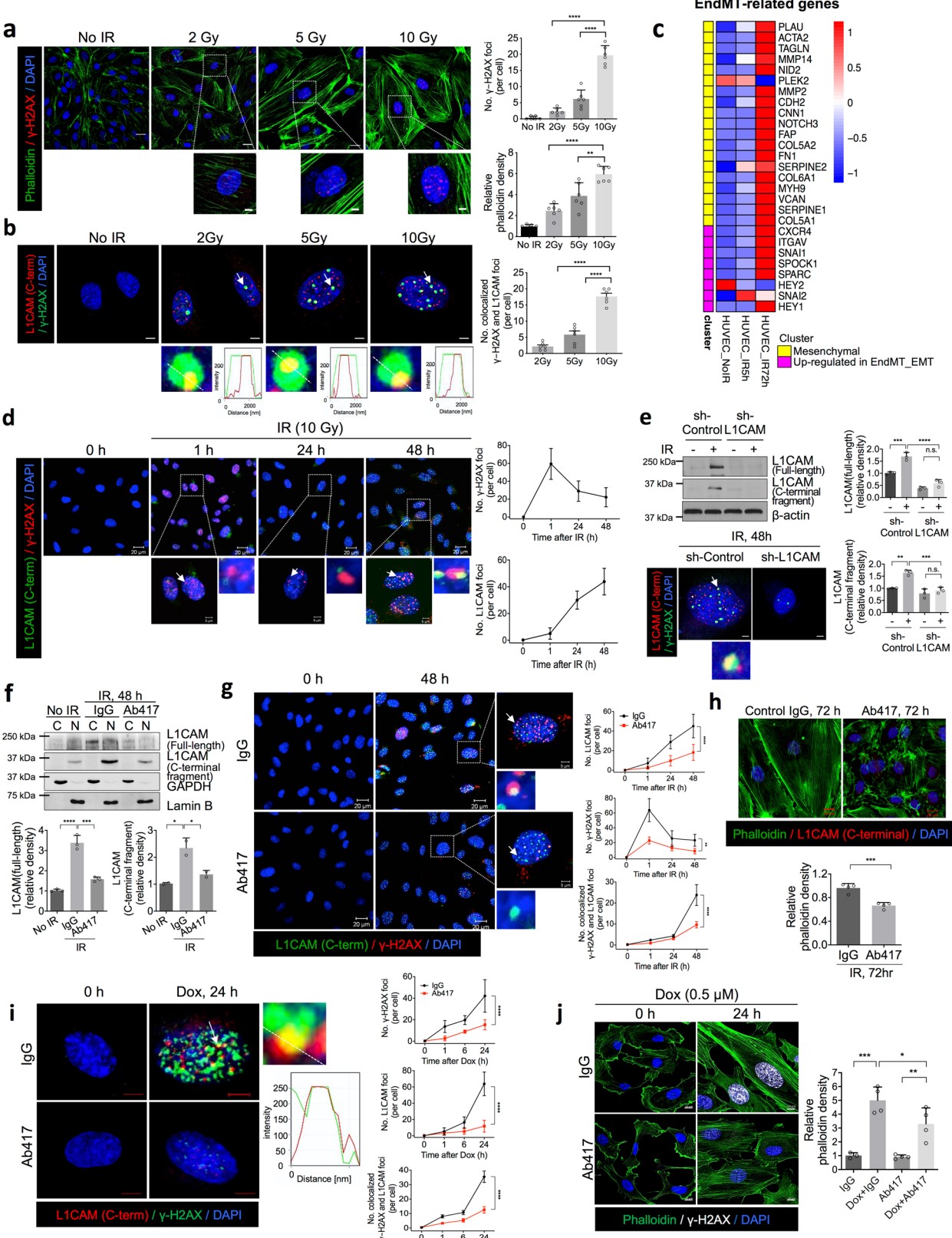

that 64 % of GFP-positive foci had a lifespan of over 48 h (Supplementary Fig 2b).

Nuclear L1-CT foci were detected by the binding of an anti-L1CAM antibody to the C-terminal epitope but not to the N-terminal epitope (Fig. 1b, d), which demonstrates that the nuclear L1-CT foci were derived from L1-CT cleaved from L1CAM.

Altogether, we suggest that nuclear localisation of L-CT is correlated with persistent DNA damage, accompanied with radiation-induced EndMT.

Next, we investigated the effect of nuclear L1-CT on radiation-induced persistent DNA damage in vascular ECs. The expression levels of intact L1CAM (250 kDa) and L1-CT

**Fig. 1 L1-CT fragments colocalize with persistent γ-H2AX foci after irradiation (IR) or Dox treatment in HUVECs. a, b** Immunofluorescence staining and quantification of phalloidin, γ-H2AX and L1-CT 48 h post IR (2 Gy, 5 Gy, or 10 Gy) in HUVECs (left panels, magnification, ×400). Scale bar = 20 μm (enlarged, 5 μm). Bar graphs quantifying the number of γ-H2AX foci, the phalloidin density, and the number of colocalized foci (right panels). For quantification of phalloidin density, error bars represent mean ± SEM (2 Gy vs. 10 Gy $p < 0.0001$; 5 Gy vs. 10 Gy $p = 0.0015$). For quantification of the number of γ-H2AX foci, error bars represent mean ± SD (****$p < 0.0001$). Colocalized foci (marked by white arrow) are amplified and graphs represent quantification of L1-CT and γ-H2AX signals in selected regions of dotted lines (bottom panels). Error bars represented mean ± SEM (****$p < 0.0001$). **c** Heat ma*p* of the RNA-seq analysis results showing radiation-induced EndMT and the mesenchymal phenotype. Total RNA was isolated from HUVECs before and after 10 Gy IR (5 h, 72 h). **d** Immunofluorescence staining of L1-CT and γ-H2AX in HUVECs at 0, 1, 24, and 48 h post IR (10 Gy; left panels). Quantification of γ-H2AX foci and nuclear L1CAM (magnification, ×400; right panels). The number of γ-H2AX foci with an intensity greater than 40 and a foci diameter of 0.1 μm was counted. Scale bar = 20 μm (enlarged, 5 μm). **e** Immunoblotting and quantification of full-length L1CAM and L1-CT fragments from HUVECs transfected with lentiviral shRNA targeting L1CAM 48 h after IR (10 Gy; upper panels). Error bars represent mean ± SD (full-length L1CAM: sh-Control IR − vs. + $p = 0.0003$; sh-Control IR + vs. sh-L1CAM IR + $p < 0.0001$, L1-CT fragments: sh-Control IR − vs. + $p = 0.0013$; sh-Control IR + vs. sh-L1CAM IR + $p = 0.0006$). Immunofluorescence staining of γ-H2AX and L1-CT 48 h post IR (10 Gy) in HUVECs (magnification, ×400). Scale bar = 5 μm. **f** Immunoblotting (upper panel) of full-length L1CAM and L1-CT fragments in the cytoplasmic (C) and nuclear (N) fractions of HUVECs 48 h post IR (10 Gy). Quantification of full-length L1CAM in the cytoplasmic fractions and L1-CT fragments in the nuclear fractions. HUVECs were treated with control IgG or Ab417 (20 μg/mL) before IR. GAPDH and lamin B were used as cytoplasmic and nuclear markers, respectively. Error bars represent mean ± SD from independent experiments (full-length L1CAM: No IR vs. IR + IgG $p < 0.0001$; IR + IgG vs. IR + Ab417 $p = 0.0001$, L1-CT fragments: No IR vs. IR + IgG $p = 0.0228$; IR + IgG vs. IR + Ab417 $p = 0.0448$). **g** Immunofluorescence staining for L1-CT and γ-H2AX 0 and 48 h post IR (10 Gy) in HUVECs pre-treated with control IgG or Ab417 (20 μg/mL) (upper panel). Quantification of colocalization of γ-H2AX foci with L1CAM (magnification, ×400) (lower panel). Scale bar = 20 μm (enlarged, 5 μm). **h** Immunofluorescence staining (upper panel) and quantification (lower panel) of phalloidin and L1-CT 72 h post IR (10 Gy) in HUVECs pre-treated with control IgG or Ab417 (20 μg/mL; magnification, ×400). Scale bar = 20 μm. Error bars represent mean ± SD ($p = 0.0007$). **i** Immunofluorescence staining for L1-CT and γ-H2AX at 0 and 24 h after Dox treatment in HUVECs pre-treated with control IgG or Ab417 (20 μg/mL; left panel). Colocalized foci (marked by white arrow) are amplified and graphs represent quantification of L1-CT and γ-H2AX signals in selected regions of dotted lines (middle panels). Quantification of colocalization of γ-H2AX foci with L1CAM (magnification, ×400; right panel). Scale bar = 5 μm. **j** Immunofluorescence staining and quantification of phalloidin and γ-H2AX 0 and 24 h after Dox treatment in HUVECs pre-treated with control IgG or Ab417 (magnification, ×400). Scale bar = 10 μm. Error bars represent mean ± SD (IgG vs. Dox + IgG $p = 0.0002$; Dox + IgG vs. Dox + Ab417 $p = 0.0482$; Ab417 vs. Dox + Ab417 $p = 0.0065$). For quantification of γ-H2AX foci and γ-H2AX foci colocalized with L1CAM, the foci in each sample were counted at least 70 cells per field (magnification, ×100). The average number of foci/cell was determined from >6 fields (magnification, ×100). Data are representative of three independent experiments. (**h**: two-tailed Student's *t*-test, all other panels: one-way ANOVA for multiple comparisons).

fragments (32–37 kDa) were higher in 10 Gy irradiation-treated cells 48 h post-irradiation than in untreated cells and were lower in L1CAM lentiviral shRNA-transfected cells than in control shRNA-transfected cells 48 h post-irradiation (Fig. 1e). According to immunofluorescence analysis, we used the lentiviral shRNA to knockdown L1CAM to show that the the L1-CT and γ-H2AX colocalization disappeared, indicating that this colocalization was dependent on L1CAM expression. (Fig. 1e, bottom panel). Similarly, higher expression of cytosolic L1CAM and nuclear L1-CT was observed in irradiated HUVECs than in untreated cells, but Ab417 treatment attenuated this radiation-mediated increase in expression 48 h after irradiation (Fig. 1f). These results were confirmed by immunofluorescence showing that Ab417 efficiently decreased the nuclear localisation of L1-CT in HUVEC-irradiated cells (Fig. 1g). Importantly, this was accompanied by a reduction in the number of γ-H2AX foci and, consequently, the number of colocalized L1-CT and γ-H2AX foci (Fig. 1g). Furthermore, Ab417 pre-treatment resulted in a significant reduction in the elongated fibrotic phenotype of HUVECs 72 h after irradiation, showing reduced size of irradiated ECs with flattened nuclei compared to non-irradiated ECs (Fig. 1h and Supplementary Fig. 2c). Although Ab417 did not significantly affect radiation-induced cell death, it significantly increased the proliferation rate and reduced the aneuploid cell population 120 h after irradiation (Supplementary Fig. 2d, e), indicating that persistent DNA damage in ECs may be related to aneuploidy and a morphological change from normal to fibrotic ECs. Additionally, we found that in prolonged culture (up to 40 days following irradiation), Ab417-treated ECs prominently sustained reduced DNA damage and aneuploidy compared to IgG-treated ECs, The number and intensity of L1-CT foci were also decreased at longer time points in Ab417-treated cells, compared to IgG-treated ECs (Supplementary Fig. 2f). These

results suggest that Ab417 regulates radiation-induced persistent DNA damage and the fibrotic changes in ECs.

Finally, we examined whether L1CAM can also regulate Dox-induced DNA damage in HUVECs. As expected, γ-H2AX foci were detected 1 h after Dox treatment and increased in number after 6 h (Supplementary Fig. 2g), confirming that Dox induces persistent DNA damage. L1-CT colocalized with γ-H2AX foci 24 h after Dox treatment, which was significantly reduced by Ab417 treatment (Fig. 1i). Ab417 also did not significantly affect Dox-induced cell death, it significantly increased the proliferation (Supplementary Fig. 2h). Dox treatment also enhanced fibrotic cytoskeletal development (Fig. 1j); and increased L1CAM, α-SMA, VCAM-1 and γ-H2AX expression (Supplementary Fig. 2i), and these effects were inhibited by Ab417 treatment (Fig. 1j and Supplementary Fig 2i).

Taken together, our findings suggest that nuclear L1-CT regulates radiation- and Dox-induced persistent DNA damage via colocalization with γ-H2AX foci in vascular ECs with a fibrotic phenotype.

**Nuclear L1CAM foci directly regulate persistent DNA damage repair in ECs.** Next, we studied the precise role of nuclear L1-CT in the DNA damage response in vascular ECs. First, we studied if L1CAM cleavage was necessary for and L1-CT and γ-H2AX foci colocalization. To this end we used the inhibitor L-685,458, which is known to inhibit L1CAM cleavage by presenilin (γ-secretase)[41,42], and found that inhibition of the γ-secretase in irradiated HUVECs resulted in decreased radiation-induced L1CAM overexpression, L1-CT levels, and L1-CT and γ-H2AX foci colocalization (Fig. 2a).

Next, to examine the functional role of cleaved nuclear L1-CT, we generated an L1-4A mutant construct with a mutation in the previously described L1CAM nuclear localisation signal (NLS)[43,44] and an endocytosis-deficient L1CAM mutant lacking

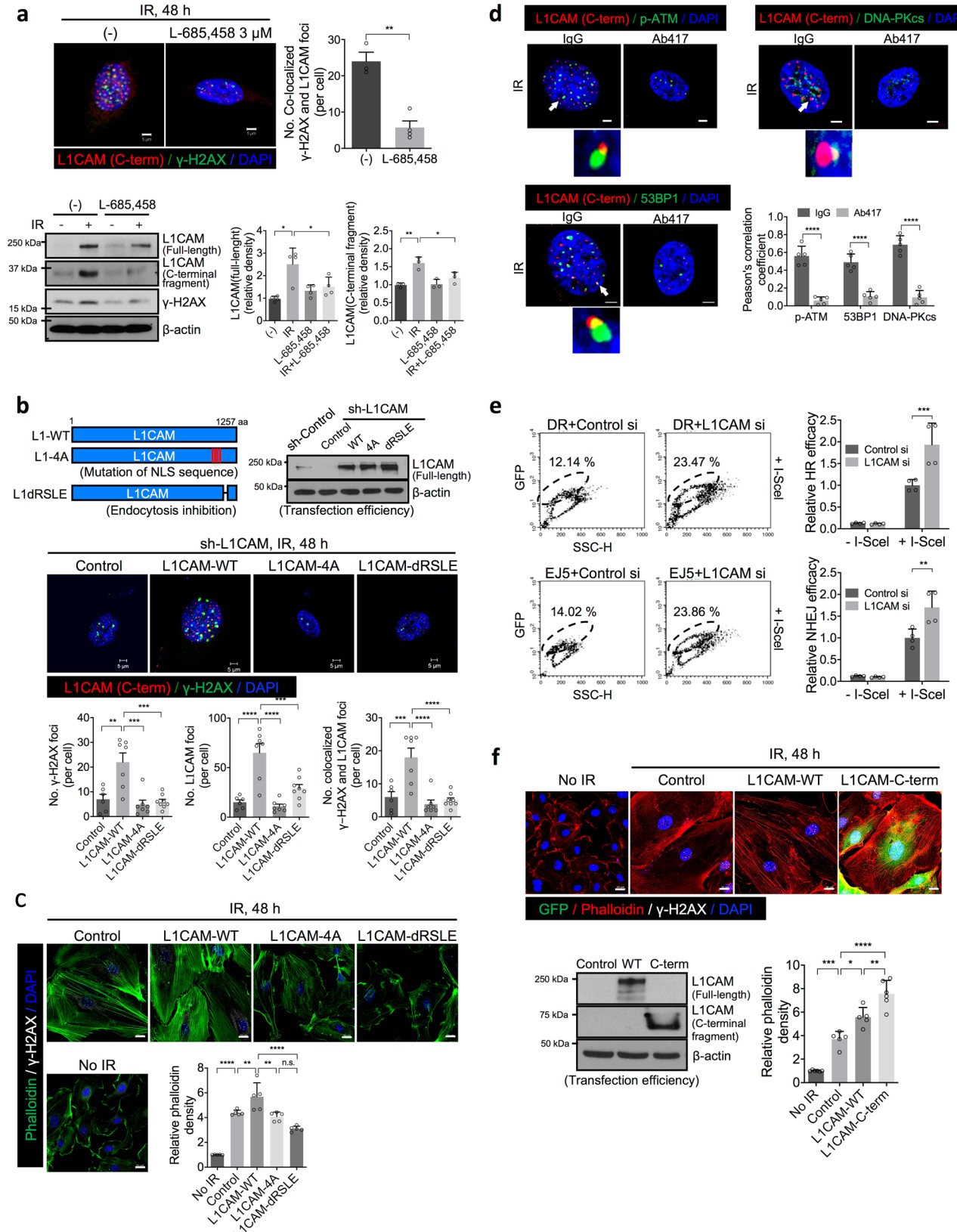

the RSLE motif (L1-dRSLE) (Fig. 2b), which is involved in clathrin-mediated L1CAM endocytosis and cleavage[45]. To minimise the effect of endogenous wild-type L1CAM, we suppressed L1CAM expression using lentivirus-expressed shRNA specific for L1CAM. At 48 h post-irradiation, compared with L1-WT-expressing HUVECs, L1-4A- or L1-dRSLE-overexpressing

HUVECs showed a significant reduction in the number of L1-CT foci and this was accompanied by a significant reduction in numbers of γ-H2AX foci and, consequently, in the number of colocalized foci (Fig. 2b). Moreover, HUVECs expressing L1-4A and L1-dRSLE did not show the radiation-induced fibrotic changes observed in L1CAM-expressing HUVECs (Fig. 2c).

**Fig. 2 Nuclear localisation of L1CAM inhibits γ-H2AX foci resolution and DNA repair in HUVECs. a** Immunofluorescence staining and quantification of γ-H2AX foci colocalized with L1-CT (upper left panel; magnification, ×400) and immunoblotting (lower left panel) and quantification (right panels) of full-length L1CAM, L1-CT fragments, and γ-H2AX at 48 h post-irradiation (10 Gy) in HUVECs with or without γ-secretase inhibitor L-685,458 (3 μM) treatment. For quantification of γ-H2AX foci colocalized with L1-CT, error bars represent mean ± SEM ($p = 0.0017$). For quantification of full-length L1CAM and L1-CT fragments, error bars represent mean ± SD (full-length L1CAM: (-) vs. IR $p = 0.0011$; IR vs. IR + L-685,458 $p = 0.0233$, L1-CT fragments: (-) vs. IR $p = 0.0017$, IR vs. IR + L-685,458 $p = 0.0177$). **b**, **c** HUVECs were transfected with human full-length (L1-WT), NLS-mutated (L1-4A), and endocytosis-deficient (L1-dRSLE) L1CAM vectors after knockdown of endogenous L1CAM. **b** Scheme of L1-WT, L1-4A and L1-dRSLE constructs (top left panel). Immunoblotting of full-length L1CAM in HUVECs (top right panel) and immunofluorescence staining (middle panels) and quantification (bottom panels) of L1-CT and γ-H2AX in HUVECs 48 h post-irradiation (10 Gy; magnification, ×400). Scale bar = 5 μm. Error bars represent mean ± SEM (No. γ-H2AX foci: Control vs. L1CAM-WT $p = 0.001$; L1CAM-WT vs. L1CAM-4A $p = 0.0001$; L1CAM-WT vs. L1CAM-dRSLE $p = 0.0002$, No. L1-CT foci: ****$p < 0.0001$; L1CAM-WT vs. L1CAM-dRSLE $p = 0.0004$, No. colocalized foci: Control vs. L1CAM-WT $p = 0.006$; ****$p < 0.0001$). **c** Immunofluorescence staining and quantification of phalloidin and γ-H2AX in HUVECs 48 h post-irradiation (10 Gy; magnification, ×400). Scale bar = 20 μm. Error bars represent mean ± SD (Control vs. L1CAM-WT $p = 0.0096$; L1CAM-WT vs. L1CAM-4A $p = 0.0014$; ****$p < 0.0001$). **d** Immunofluorescence staining and pearson's correlation coefficient of L1-CT colocalized with p-ATM, 53BP1, and DNA-PKcs in the nuclei of Ab417-pre-treated HUVECs 48 h post-irradiation (10 Gy magnification, ×400). Scale bar = 5 μm. Error bars represent mean ± SD (****$p < 0.0001$). **e** Flow cytometry analysis of cells with GFP positivity resulting from DNA repair and quantification of HR (upper panel) and NHEJ (lower panel) efficiency in L1CAM-knockdown HUVECs. The HUVECs were transiently transfected with the DR-GFP or EJ5-GFP construct along with control or L1CAM siRNA and were then transfected with a SceI plasmid to induce DNA damage. Error bars represent mean ± SD (HR efficacy $p = 0.0005$, NHEJ efficacy $p = 0.0011$). **f** Immunofluorescence staining (upper panel) for GFP, phalloidin, γ-H2AX in HUVECs 48 h post-irradiation (10 Gy). Quantification of phalloidin is shown (magnification, ×400; right panel). Scale bar = 20 μm. Error bars represent mean ± SD (No IR vs. IR + Control $p = 0.0001$; IR + Control vs. IR + L1CAM-WT $p = 0.0063$; IR + L1CAM-WT vs. IR + L1CAM-C-term $p = 0.0027$, IR + Control vs. IR + L1CAM-C-term $p < 0.0001$). HUVECs were transfected with L1-WT or L1-CT lentiviral vectors tagged with N-terminal GFP and C-terminal His. Transfection efficiency (left panel) was tested by immunoblotting for L1-WT and L1-CT. For quantification of foci colocalized with L1CAM, the colocalized foci in each sample were counted in a minimum of 70 cells per field (magnification, ×100). The average number of colocalized foci/cell was determined from five fields (magnification, ×100). The data are presented the means ± SDs and ±SEMs from three independent experiments. (**a** upper left panel: two-tailed Student's *t*-test; **a** all other panels and **b**, **c**, **f**: one-way ANOVA for multiple comparisons; **d**, **e** two-way ANOVA for multiple comparisons).

Consistent with the fibrotic changes observed with phalloidin staining, overexpression of L1-WT, but not L1-4A or L1-dRSLE, significantly increased radiation-induced collagen type 1 (COL1A1) and α-SMA expression above the levels observed in the irradiated controls (Supplementary Fig. 3a). From these results, we suggest that L1CAM cleavage regulates the DNA damage via nuclear L1-CT and γ-H2AX foci colocalization and the fibrotic changes in irradiated ECs.

We then aimed to determine whether nuclear L1-CT directly regulates the DNA damage response in vascular ECs. To this end first we investigated whether L1-CT foci colocalized with DNA repair enzymes, namely, the homologous recombination (HR) markers phospho-ATM (p-ATM) and 53BP1 and the non-homologous end joining (NHEJ) marker DNA-PKcs, following irradiation. Indeed we detected colocalization of L1-CT foci with p-ATM, 53BP1, and DNA-PKcs at 48 h post-irradiation in control cells treated with IgG and this colocalization was significantly reduced in cells pre-treated with Ab417 (Fig. 2d and Supplementary Fig. 3b). To determine whether L1-CT is recruited to the targeted DNA damage site, we studied L1-CT and γ-H2AX foci colocalization in HUVECs after causing site-specific DNA damage using laser microirradiation and found that L1-CT was recruited to the γ-H2AX foci 20 min after microirradiation (Supplementary Fig. 3c). We next investigated whether L1-CT directly regulates double-strand break repair via HR or NHEJ using the previously described DR-GFP and EJ5-GFp reporters of HR and NHEJ, respectively[46–48]. In these reporters, the *GFP* genes contain a cut sites for the I-SceI nuclease, whose activity causes a double-strand break; repair of the double-strand break by HR and NHEJ restores the DR-GFP and EJ5-GFP genes, respectively, resulting in functional GFP proteins. We transfected DR-GFP and EJ5-GFP into HUVECs treated with or without L1CAM-specific siRNA and found that the percentages of DR-GFP+ and EJ5-GFP+ cells among L1CAM-specific siRNA-treated cells were significantly higher (over 2- and 1.5-fold higher, respectively) than those among control siRNA-treated cells (Fig. 2e and Supplementary Fig. 3d), indicating that L1-CT is

required for both HR and NHEJ repair. As L1-CT fragments are produced via cleavage after DNA damage, we generated 32 kDa GFP-tagged L1-CT. Compared with L1CAM, GFP-tagged L1-CT increased the number of γ-H2AX foci and enhanced the transition to a fibrotic phenotype after irradiation (Fig. 2f). Consistent with these results, an immunoprecipitation assay showed that L1-CT binds to r-H2AX in irradiated ECs transfected with GFP-L1-CT (Supplementary Fig. 3e).

Collectively, the findings suggest that L1-CT is required for the DNA repair process via colocalization with γ-H2AX foci in vascular ECs.

## Ab417 promotes the endocytic internalisation and lysosomal degradation of L1CAM.

In accordance with our previous work[36], Ab417 inhibited the expression levels of full-length L1CAM and its C-terminal fragment in this study. L1CAM has also been reported to be regulated by its lysosomal degradation and endocytosis[49]. To clarify how Ab417 directly affects L1CAM levels, we examined the effects of Ab417 on the endocytosis and lysosomal degradation of L1CAM. Ab417 was labelled with pHrodo fluorescence dye, which increases fluorescence upon internalisation into acidic endosomes and lysosomes. The red fluorescence of internalised pHrodo-labelled Ab417 was observed at 1 h and peaked at 18 h after incubation in HUVECs (Supplementary Fig. 4a). Additionally, we examined whether Ab417 can directly induce the endocytotic internalisation of endogenous L1CAM, resulting in lysosomal degradation. An anti-L1CAM antibody against endogenous L1CAM was labelled with green pHrodo dye. The pHrodo Green-labelled anti-L1CAM antibody colocalized with endocytic vesicles containing Ab417 with red fluorescence (Supplementary Fig. 4a), suggesting that Ab417 induced the internalisation and endocytosis of membrane L1CAM. Ab417 significantly reduced L1CAM expression in WT-L1CAM-overexpressing HUVECs, and this effect was inhibited by treatment with the lysosomal degradation inhibitor bafilomycin-A1 (Supplementary Fig. 4b,c). These results suggest

that Ab417 promotes the internalisation, endocytosis and subsequent lysosomal degradation of L1CAM in HUVECs.

Additionally, to address the possibility of unwanted effects by Ab417, we performed an NK cell-mediated antibody-dependent cellular cytotoxicity (ADCC) assay to determine whether Ab417 on HUVECs can lead to effector function. Ab417 on HUVECs did not trigger the ADCC effect, and the ADCC effect of the anti-CD20 antibody on Raji cells was used as a positive control (Supplementary Fig. 4d). This result supports our data that Ab417 protects against EC damage following radiation and Dox treatment without ADCC effects.

Altogether, our findings suggest that Ab417 reduced L1CAM expression by the endocytotic lysosomal degradation of full-length L1CAM and subsequently reduced the expression of the L1-CT fragment. Therefore, the Ab417 antibody inhibits persistent DNA damage by reducing the expression of full-length L1CAM and the subsequent nuclear L-CT fragment and does not directly interfere with the cleavage and nuclear translocation of L1CAM.

**The anti-L1CAM antibody reduces p53 knockout- and radiation- and Dox-induced DNA damage in cardiovascular ECs.** As the results of the above subsection indicated that L1-CT regulates the DNA repair process in vascular ECs, we next investigated whether the anti-L1CAM antibody can prevent radiation-induced cardiac damage. We specifically examined cardiac damage enhanced by EC-p53 deletion, as endothelial p53 deletion has been reported to enhance radiation-induced cardiac damage[50,51]. To this end, first we examined the effect of p53 expression on L1CAM expression and nuclear localisation. HUVECs were transfected with p53-specific siRNA or control siRNA and irradiated; 48 h post-irradiation, HUVECs transfected with p53-specific siRNA showed significantly (fivefold) higher *L1CAM* mRNA levels than HUVECs transfected with control siRNA (Fig. 3a). These results were confirmed at the protein level, showing that p53 knockdown markedly increased L1-CT and full-length L1CAM expression after irradiation; and this effect was inhibited by Ab417 treatment (Fig. 3b). Moreover, immunofluorescence data showed that p53 knockdown markedly increased the number of L1-CT foci and the colocalization of these foci with γ-H2AX after irradiation, and that these effects were reversed in Ab417-treated cells (Fig. 3c and Supplementary Fig. 5a). p53 knockdown in HUVECs increased radiation-induced cell death and decreased cell proliferation reduced by irradiation, and these effects were reversed in Ab417-treated cells (Supplementary Fig 5b).

After confirming that p53 can affect L1CAM expression and translocation to the nucleus, we next investigated whether Ab417 can prevent irradiation- or Dox-induced DNA damage in p53-deficient cardiovascular ECs using EC-specific *Trp53* knockout (EC-p53KO) mice (Tie2-*Cre*;*Trp53*^flox/flox^). Compared with wild-type mice, non-irradiated EC-p53KO mice did not show any significant differences in arterial wall thickness, vascular fibrosis, microvessel density, or myocardial dilation (Supplementary Fig. 5c). By contrast, at 3 weeks after whole-heart irradiation, DNA damage, nuclear L1-CT and the colocalization of L1-CT and γ-H2AX in cardiovascular ECs of control IgG-treated EC-p53KO mice were increased compared to those of wild-type mice. Importantly, these effects in EC-p53KO irradiated mice were reversed by Ab417 treatment (Fig. 3d and Supplementary Fig. 5d, e).

In irradiated cardiac tissues of WT mice, ECs (7.1-fold) were more γ-H2AX-positive than cardiomyocytes (3.4-fold), compared to non-irradiated cardiac tissues, and these phenotypes were enhanced in the irradiated cardiac tissues of pEC-53 KO mice;

additionally, Ab417 reduced DNA damage in ECs to a greater extent than cardiomyocytes in the irradiated cardiac tissues of pEC-53 KO mice (Supplementary Fig. 5f).

Consistent with these findings, histological analyses showed that the radiation-induced increases in inflammation, vascular fibrosis, arterial wall thickness, and vessel destruction observed in EC-p53KO mice were efficiently ameliorated by Ab417 treatment (Fig. 3e). Moreover, the increase in EndMT, measured as number of α-SMA⁺CD31⁺ cells (Fig. 3f), and in hypoxic regions of cardiomyocytes (Supplementary Fig. 5g) in irradiated EC-p53KO hearts compared with irradiated wild-type hearts were attenuated by Ab417 treatment. The cardiac troponin T (cTnT) levels were also significantly lower in the heart tissues of irradiated EC-p53KO mice than in those of non-irradiated and irradiated wild-type mice, but these effects were attenuated by Ab417 (Fig. 3g). These findings indicate that EC-p53KO affects hypoxia and cardiomyocyte dysfunction and enhances radiation-induced EndMT, but these effects were reduced by Ab417 treatment.

Finally, given that Ab417 treatment ameliorated detrimental irradiation-induced effects in EC-p53KO mouse hearts, we investigated the effect of Ab417 treatment on Dox-induced toxicity in EC-p53KO mice. Five days after Dox treatment, we observed that the DNA damage and nuclear L1-CT levels in cardiovascular ECs of control IgG-treated EC-p53KO mice were significantly increased compared with those of wild-type mice; however, these effects were not observed in Ab417-treated EC-p53KO mice (Supplementary Fig. 6a). Collectively, these results indicate that the anti-L1CAM antibody efficiently inhibits the L1CAM nuclear localisation and DNA damage induced by EC-p53 deletion. Consistent with the DNA damage in ECs, EndMT and TnT loss occurred in hypoxic cardiomyocytes, but these effects were reduced by Ab417 treatment.

**Ab417 inhibits radiation- and Dox-induced cardiovascular DNA damage and EndMT.** We next investigated whether L1CAM affects cardiovascular DNA damage during radiation- and Dox-induced cardiotoxicity. Whole-heart irradiation of mice with 16 Gy specifically increased the expression of L1CAM and γ-H2AX in vascular ECs compared to that in non-irradiated ECs, and this effect was reversed by Ab417 treatment (Fig. 4a). In the irradiated hearts of control IgG-treated mice, γ-H2AX foci were prominently detected in CD31⁺ ECs and the numbers of these foci were markedly reduced after Ab417 treatment. (Supplementary Fig. 7a). However, γ-H2AX foci were much less abundant in WGA-stained cardiomyocytes or in α-SMA⁺ fibroblasts, compared to podocalyxin⁺ ECs (Supplementary Fig. 7a, b). Moreover, histopathological examination showed that heart irradiation caused vascular fibrosis around the arteriole, large vessel destruction, and reduced microvessel density, and these vascular damage patterns were significantly inhibited by Ab417 treatment (Fig. 4b). Furthermore, Ab417 treatment reduced the radiation-induced increases in inflammatory cell infiltration around the irradiated vessels (Fig. 4b). Serum levels of C-reactive protein (CRP), E-selectin, and ICAM-1 (circulating markers of vascular inflammation), which were higher in irradiated control IgG-treated mice than in non-irradiated mice, were also significantly reduced by Ab417 treatment (Fig. 4c). Antibodies against L1-CT showed enhanced specificity for vascular endothelial L1CAM in IgG-treated irradiated hearts (Fig. 4d); by contrast, antibodies against the N-terminal domain did not specifically target vascular ECs; rather, binding of these antibodies was broadly detected in the tissue, similar to the general pattern of secretory factor binding (Supplementary Fig. 7c). Notably, nuclear L1-CT was predominantly detected in cardiovascular ECs in irradiated mice but was much less abundant in Ab417-treated

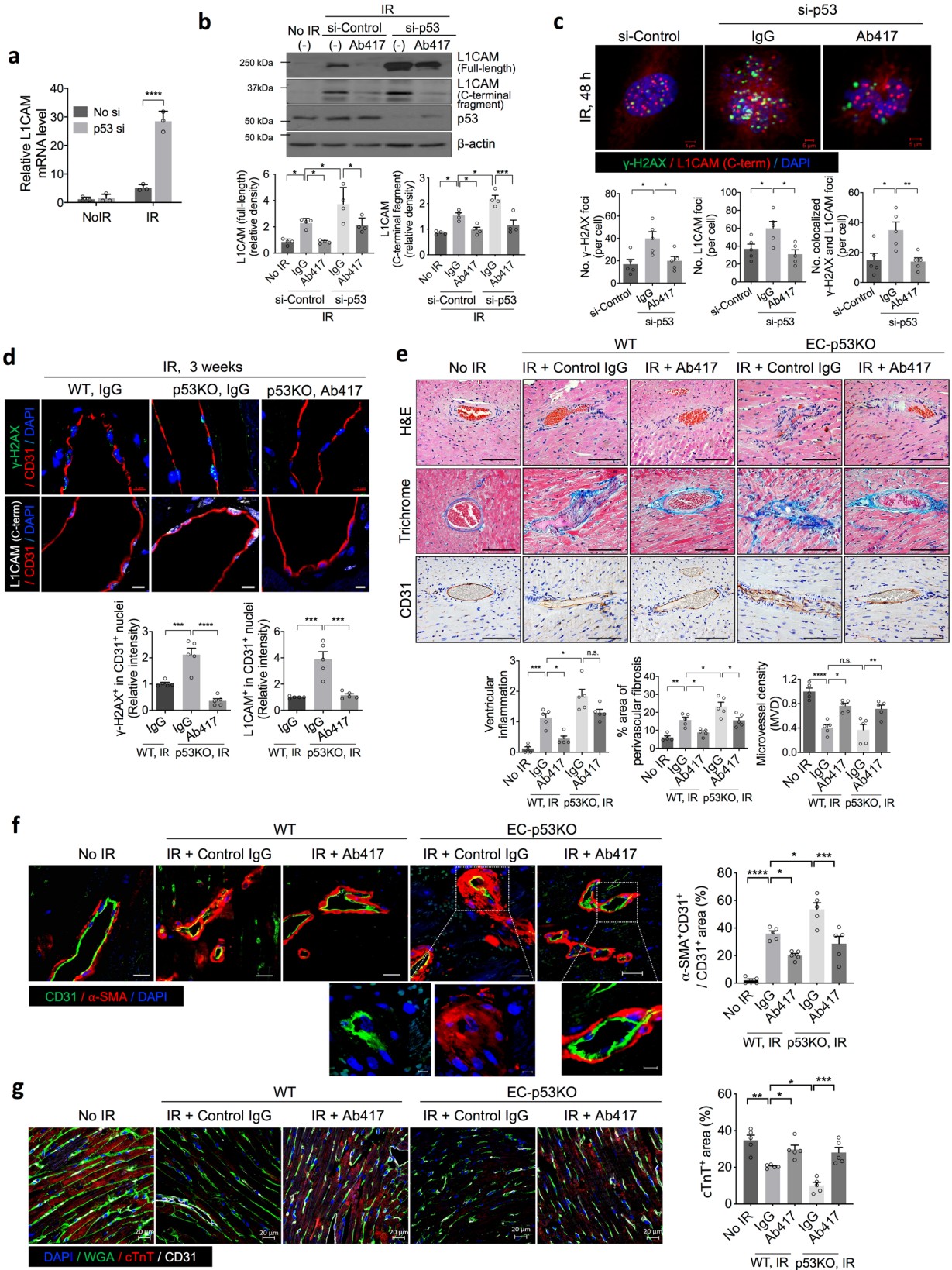

mice (Fig. 4d and Supplementary Fig. 7d). Furthermore, L1-CT was more highly expressed in ECs than in the cardiomyocytes or fibroblasts of either non-irradiated or irradiated mice (Supplementary Fig. 7d, e). EndMT and TnT loss in hypoxic cardiomyocytes occurred in irradiated mice, but these effects were reduced by Ab417 treatment (Fig. 4e and Supplementary Fig. 7f).

When similar experiments were conducted in Dox-treated mice, an induction in L1CAM expression and DNA damage in vascular ECs was observed (Fig. 4f). Additionally, compared with untreated controls, Dox-treated mice displayed increased vascular occlusion, inflammation, arterial wall thickness, and fibrosis along with a reduced microvessel density (Fig. 4g). Notably, Ab417

**Fig. 3 An anti-L1CAM antibody prevents p53 knockout- and radiation-induced DNA damage in heart ECs. a** RT-qPCR analysis of L1CAM in p53-knockdown HUVECs 48 h post-irradiation (10 Gy). Error bars represent mean ± SD from three independent experiments ($p < 0.0001$). **b** Immunoblotting (upper panel) and quantification (lower panel) of full-length L1CAM and L1-CT 48 h post-irradiation (10 Gy) in p53-knockdown HUVECs pre-treated with control IgG or Ab417 (20 μg/mL). Error bars represent mean ± SD from four independent experiments (full-length L1CAM: No IR vs. si-Control IR + IgG $p = 0.0468$; si-Control IR + IgG vs. si-Control IR + Ab417 $p = 0.0468$; si-Control IR + IgG vs. si-p53 IR + IgG $p = 0.0462$; si-p53 IR + IgG vs. si-p53 IR + Ab417 $p = 0.0218$, L1-CT: No IR vs. si-Control IR + IgG $p = 0.0124$; si-Control IR + IgG vs. si-Control IR + Ab417 $p = 0.0484$; si-Control IR + IgG vs. si-p53 IR + IgG $p = 0.0206$; si-p53 IR + IgG vs. si-p53 IR + Ab417 $p = 0.0004$). **c** Immunofluorescence staining (upper panel) and quantification (lower panel) for γ-H2AX and L1-CT. The results of quantification of γ-H2AX foci, nuclear L1CAM, and colocalization of γ-H2AX foci with L1CAM are shown (magnification, ×400). Scale bar = 5 μm. Error bars represent mean ± SEM (No. γ-H2AX foci: si-Control vs. si-p53+IgG $p = 0.0107$; si-p53+IgG vs. si-p53 + Ab417 $p = 0.0242$, No. L1-CT foci: si-Control vs. si-p53+IgG $p = 0.0425$; si-p53+IgG vs. si-p53 + Ab417 $p = 0.0119$, No. colocalized foci: si-Control vs. si-p53 + IgG $p = 0.0131$; si-p53 + IgG vs. si-p53 + Ab417 $p = 0.0097$). For quantification of foci colocalized with L1CAM, the colocalized foci in each sample were counted in at least 70 cells per field (magnification, ×100, $n = 5$). **d**, **e** Wild-type or EC-p53KO mice were injected intravenously with control IgG or Ab417 (10 mg/kg) and subjected to 17.5 Gy thoracic irradiation ($n = 5$ animals per group). **d** Immunofluorescence staining (upper panels) for γ-H2AX, L1CAM, and CD31 in heart tissues 3 weeks post-irradiation and quantification (lower panels) of γ-H2AX⁺ cells and nuclear L1CAM⁺ cells among CD31⁺ cells (magnification, ×400). Scale bar = 5 μm. Error bars represent mean ± SEM (γ-H2AX⁺ in CD31⁺ nuclei: WT⁺IR vs. p53KO+IR $p = 0.0005$; p53KO + IgG vs. p53KO + Ab417 $p < 0.0001$, L1CAM⁺ in CD31⁺ nuclei: WT⁺IR vs. p53KO+IR $p = 0.0001$; p53KO+IgG vs. p53KO + Ab417 $p = 0.0002$). **e** Haematoxylin and eosin staining, Masson's trichrome staining, and immunohistochemical detection of CD31 in heart tissues 3 weeks post-irradiation (upper panels) and quantification of ventricular inflammation, perivascular fibrosis area, and microvessel density per field (magnification, ×200; lower panels). Scale bar = 100 μm. Error bars represent mean ± SEM (ventricular inflammation: No IR vs. WT IR + IgG $p = 0.0003$; WT IR + IgG vs. WT IR + Ab417 $p = 0.0125$; WT IR + IgG vs. p53KO IR + IgG $p = 0.0104$, perivascular fibrosis area: No IR vs. WT IR + IgG $p = 0.0021$; WT IR + IgG vs. WT IR + Ab417 $p = 0.0328$; WT IR + IgG vs. p53KO IR + IgG $p = 0.0203$; WT IR + IgG vs. WT IR + Ab417 $p = 0.0165$, MVD: No IR vs. WT IR + IgG $p < 0.0001$; WT IR + IgG vs. WT IR + Ab417 $p = 0.0055$; WT IR + IgG vs. WT IR + Ab417 $p = 0.0068$). **f** Immunofluorescence staining (left panels) of α-SMA and CD31 in heart tissues 3 weeks post-irradiation ($n = 5$ animals per group) and quantification (right panels) of the α-SMA⁺CD31⁺ area in the CD31⁺ area (magnification, ×400). Scale bar = 20 μm. Error bars represent mean ± SEM (No IR vs. WT IR + IgG $p < 0.0001$; WT IR + IgG vs. WT IR + Ab417 $p = 0.0251$; WT IR + IgG vs. p53KO IR + IgG $p = 0.011$; p53KO IR + IgG vs. p53KO IR + Ab417 $p = 0.0004$). **g** Immunofluorescence staining (left panel) of WGA, cTnT, and CD31 ($n = 5$ animals per group) and quantification (right panel) of the cTnT area per field (magnification, ×400). Scale bar = 20 μm. Error bars represent mean ± SEM (No IR vs. WT IR + IgG $p = 0.0014$; WT IR + IgG vs. WT IR + Ab417 $p = 0.0489$; WT IR + IgG vs. p53KO IR + IgG $p = 0.0253$; p53KO IR + IgG vs. p53KO IR + Ab417 $p = 0.001$). *$p < 0.05$, **$p < 0.01$, ***$p < 0.001$, ****$p < 0.0001$, ns: not significant (**a** Two-way ANOVA for multiple comparisons; all other panels: one-way ANOVA for multiple comparisons).

treatment significantly inhibited this Dox-induced cardiac damage. In addition, Ab417 treatment significantly lowered Dox-induced serum CRP and E-selectin levels but the not serum ICAM-1 level (Fig. 4h). Similar to results observed in irradiated cardiac vessels, Dox-treated cardiovascular ECs showed prominent nuclear L1-CT localisation, which was prevented by Ab417 treatment (Fig. 4i). Dox also induced vascular EndMT and TnT loss in hypoxic cardiomyocytes, and the effects were prevented by Ab417 treatment (Fig. 4j and Supplementary Fig. 7g).

Taken together, our findings suggest that Ab417 can prevent the vascular endothelial damage occurring during radiation- and Dox-induced cardiotoxicity.

**The anti-L1CAM antibody mitigates radiation- and Dox-induced cardiotoxicity.** To assess whether irradiated ECs and endothelial L1CAM can affect cardiomyocyte damage, induced pluripotent stem cell-derived cardiomyocytes (iPSC-CMs) were co-cultured with irradiated ECs with a fibrotic phenotype or with non-irradiated ECs (Fig. 5a, upper panel). Compared to co-culture with non-irradiated ECs, co-culture with irradiated ECs significantly reduced the iPSC-CM beating rate and this effect was reversed when irradiated ECs transfected with L1CAM-specific siRNA were used (Fig. 5a, Supplementary Fig. 8a and Supplementary Movies 1–3). In addition, the cTnT levels were significantly decreased in iPSC-CMs co-cultured with irradiated ECs but not in iPSC-CMs co-cultured with L1CAM-knockdown irradiated ECs (Fig. 5b, top). In the absence of co-culture, TnT expression was detected in myocytes, but not in irradiated ECs (Supplementary Fig. 8b). We examined whether the loss of TnT in cardiomyocytes was caused by cardiomyocyte death using TUNEL staining (Supplementary Fig. 8c). Apoptotic cells were not detected in cardiomyocytes but were detected in 7.5 % of irradiated ECs, indicating that TnT loss is not dependent on cell death.

Moreover, compared to co-culture with non-irradiated ECs, co-culture with irradiated ECs promoted collagen deposition on cardiomyocytes, while co-culture with L1CAM-knockdown irradiated ECs resulted in low collagen deposition levels on cardiomyocytes similar to those on cardiomyocytes incubated with non-irradiated ECs (Fig. 5b, bottom). Irradiated ECs showed higher collagen 1 expression than cardiomyocytes or non-irradiated ECs and collagen 1 secreted from irradiated ECs may increase collagen deposition on cardiomyocytes (Supplementary Fig. 6d). Overall these results suggest that irradiated ECs can directly affect cardiomyocyte function and that L1CAM expression in irradiated ECs is required for this effect.

To better define how irradiated ECs affect cardiomyocyte dysfunction, 10 Gy-irradiated human adult cardiomyocytes were cultured with conditioned medium from non-irradiated and 10 Gy-irradiated ECs (Supplementary Fig. 8e). Conditioned medium from non-irradiated ECs significantly increased the expression levels of brain natriuretic peptide (BNP) and cardiac troponin I (TNI) in irradiated human adult cardiomyocytes, but conditioned media from irradiated ECs did not (Supplementary Fig. 8e), suggesting that secretory factors from ECs may affect irradiated cardiomyocyte dysfunction following irradiation. It has been reported that Neuregulin-1 (NRG-1), a angiocrine secreted from cardiac EC, plays a critical role in the protection of cardiomyocytes[52,53]. NRG-1 increased BNP and TNI expression in irradiated cardiomyocytes (Supplementary Fig. 8e). Coincidently, endothelial-specific NRG-1 expression in irradiated hearts was prominently decreased compared to that in non-irradiated hearts, whereas Ab417-treated irradiated hearts did not exhibit decreased NRG expression (Supplementary Fig. 8f). Furthermore, RNA-seq analysis of Fig. 1c showed that the expression of *NRG-1* was significantly reduced in 10 Gy-irradiated cells 72 h after irradiation compared to 10 Gy-irradiated cells 5 h after irradiation or in non-

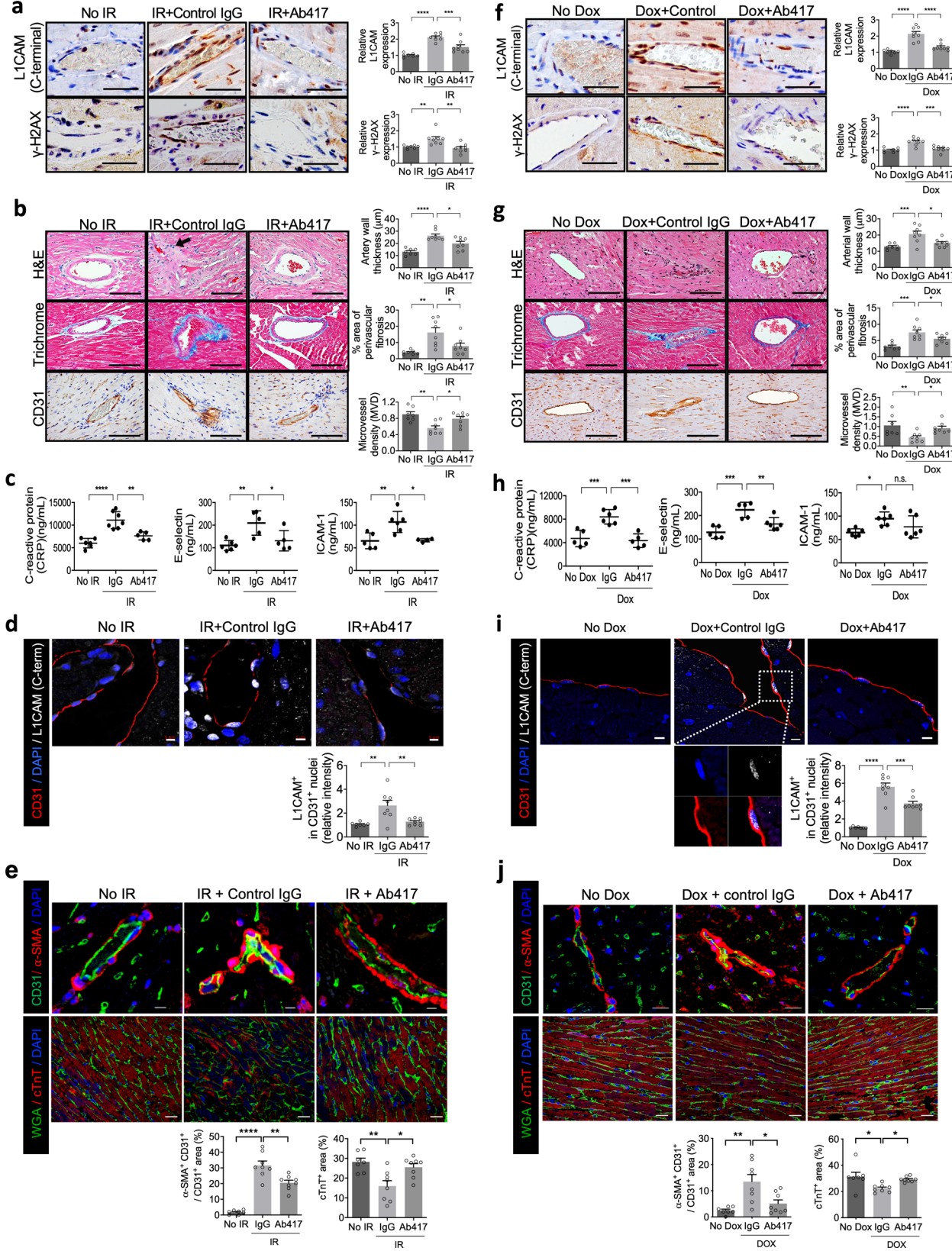

irradiated cells (Supplementary Fig. 8g). From these results, we suggest that irradiated ECs reduce cardiac angiocrine NRG-1 to protect against irradiated cardiomyocyte dysfunction, whereas Ab417 treatment can restore this reduced angiocrine function.

Considering the above results, we next investigated whether radiation- and Dox-induced cardiotoxicity can be mitigated by

the anti-L1CAM mAb. Compared with the survival of IgG-treated mice, the survival of Ab417-treated mice was prolonged after whole-heart irradiation with 16 Gy (Fig. 5c). Echocardiography showed that Ab417 markedly prevented radiation-induced LV dysfunction, as the FS and left ventricular ejection fraction (LVEF) were higher in Ab417-treated mice than in

**Fig. 4 An anti-L1CAM antibody (Ab417) inhibits irradiation (IR)- and Dox-induced vascular DNA damage and perivascular fibrosis with L1CAM nuclear localisation, EndMT and cardiac TnT loss. a–e** Mice were injected intravenously with control IgG or Ab417 (10 mg/kg) three times a week for 2 weeks and received 16 Gy thoracic IR (No IR $n = 7$; IR + IgG $n = 8$; IR + Ab417 $n = 8$). **f–j** Mice were injected intravenously with control IgG or Ab417 (10 mg/kg) with or without intraperitoneal Dox injection (4 mg/kg) three times a week for 2 weeks (No Dox $n = 7$; Dox+IgG $n = 8$; Dox+Ab417 $n = 8$). **a, f** Immunohistochemical detection (left panels) of L1-CT and γ-H2AX in heart tissues 1 week post IR (**a**) and 2 weeks after Dox treatment (**f**). The quantified L1-CT$^+$ cells and γ-H2AX$^+$ cells among ECs are shown (magnification, ×200; right panels). Scale bar = 50 μm. Error bars represent mean ± SEM (L1-CT: IR + IgG vs. IR + Ab417 $p = 0.0004$, γ-H2AX: No IR vs. IR + IgG $p = 0.0076$; IR + IgG vs. IR + Ab417 $p = 0.0017$; Dox + IgG vs. Dox+Ab417 $p = 0.0003$, **** $p < 0.0001$). **b, g** Haematoxylin and eosin staining, Masson's trichrome staining, and immunohistochemical detection of CD31 in heart tissues (left panels); and quantification of arterial wall thickness, perivascular fibrosis area, and microvessel density per field in heart tissues (magnification, ×200; right panels). Scale bar = 100 μm. The arrow in **b** indicates inflammatory cell infiltration. Error bars represent mean ± SEM (Arterial wall thickness: No IR vs. IR + IgG $p < 0.0001$; IR + IgG vs. IR + Ab417 $p = 0.0159$; No Dox vs. Dox $p = 0.0009$; Dox + IgG vs. Dox + Ab417 $p = 0.0117$, perivascular fibrosis area: No IR vs. IR + IgG $p = 0.0011$; IR + IgG vs. IR + Ab417 $p = 0.0168$; No Dox vs. Dox $p = 0.0001$; Dox + IgG vs. Dox + Ab417 $p = 0.0492$, MVD: No IR vs. IR + IgG $p = 0.0012$; IR + IgG vs. IR + Ab417 $p = 0.0184$; No Dox vs. Dox $p = 0.0093$; Dox + IgG vs. Dox + Ab417 $p = 0.0378$). **c, h** Serum CRP, E-selectin, and ICAM-1 levels 1 week post IR (**c**) and 1 week after Dox treatment (**h**). Error bars represent mean ± SD (CRP: No IR vs. IR + IgG $p < 0.0001$; IR + IgG vs. IR + Ab417 $p = 0.0015$; No Dox vs. Dox $p = 0.0007$; Dox+IgG vs. Dox+Ab417 $p = 0.0003$, E-selectin: No IR vs. IR + IgG $p = 0.0031$; IR + IgG vs. IR + Ab417 $p = 0.02$; No Dox vs. Dox $p = 0.0003$; Dox+IgG vs. Dox+Ab417 $p = 0.0082$, ICAM-1: No IR vs. IR + IgG $p = 0.0053$; IR + IgG vs. IR + Ab417 $p = 0.01$; No Dox vs. Dox $p = 0.0154$). **d, i** Immunofluorescence detection (upper panel) of L1-CT and CD31 in mouse heart tissues 1 week post IR (**d**) and 2 weeks after Dox treatment (**i**) (magnification, ×400). Scale bar = 5 μm. Quantification of L1CAM in CD31 nuclei (lower panel). Error bars represent mean ± SEM (No IR vs. IR + IgG $p = 0.0012$; IR + IgG vs. IR + Ab417 $p = 0.0033$; No Dox vs. Dox $p < 0.0001$; Dox IgG vs. Dox+Ab417 $p = 0.0004$). **e, j** Immunofluorescence staining of α-SMA and CD31 (scale bar = 10 μm) and of WGA and cTnT (scale bar = 20 μm) in heart tissues 1 week post IR (**e**) and 2 weeks post Dox treatment (**j**) (magnification, ×400; upper panels). Quantification of the α-SMA$^+$CD31$^+$ area in the CD31$^+$ area and quantification of the cTnT area per field (lower panels). Error bars represent mean ± SEM (SMA$^+$CD31$^+$ area in the CD31$^+$ area: No IR vs. IR + IgG $p < 0.0001$; IR + IgG vs. IR + Ab417 $p = 0.0011$; No Dox vs. Dox $p = 0.0013$; Dox + IgG vs. Dox + Ab417 $p = 0.0103$, cTnT area: No IR vs. IR + IgG $p = 0.0019$; IR + IgG vs. IR + Ab417 $p = 0.0107$; No Dox vs. Dox $p = 0.0126$; Dox + IgG vs. Dox + Ab417 $p = 0.0494$, one-way ANOVA for multiple comparisons). Data are representative of three independent experiments.

IgG-treated mice. However, left ventricular end systolic volume (LVESV) and left ventricular end diastolic volume (LVEDV) were lower in Ab417-treated mice than in IgG-treated mice (Fig. 5d). In addition, Ab417 significantly prevented radiation-induced myocardial dilation, and the cross-sectional area in the Ab417-treated group was similar to that observed in non-irradiated myocardium (Fig. 5e). Similar to the results observed in irradiated mice, Ab417 treatment markedly prolonged the 50% survival rate to 100 days after Dox treatment, whereas all mice treated with IgG and Dox died within 40 days after Dox treatment (Fig. 5f). Ab417 also prevented the Dox-induced decrease in LVEF (Fig. 5g). Finally, compared with untreated control mice, Dox-treated mice showed significantly reduced cardiomyocyte diameters; however, Ab417 treatment almost completely prevented this decrease (Fig. 5h). Collectively, these results suggest that irradiated ECs directly induce cardiomyocyte damage and Ab417 can prevent radiation- and Dox-induced cardiotoxicity.

**The anti-L1CAM antibody enhances radiation- and Dox-induced anti-tumour effects.** To further validate the effects of the anti-L1CAM antibody on tumours, we investigated the effects of Ab417 treatment and radiotherapy or Dox treatment on tumour growth in nude mice bearing metastatic breast carcinoma MDA-MB-231 xenografts. Compared to that in non-treated tumours, L1CAM expression significantly increased in MDA-MB-231 tumours treated with either irradiation or Dox (Supplementary Fig. 9a, b). Compared to IgG control mice, Ab417-treated mice exhibited approximately 41% tumour growth inhibition at 31 days after irradiation (Supplementary Fig. 9a). Additionally, Ab417 treatment significantly enhanced Dox-induced anti-tumour effects (Supplementary Fig. 9b). These results indicate that Ab417 increases anti-tumour effects in combination with radiation therapy and Dox treatment.

Additionally, to address whether the effects of L1CAM on cancer cells differed from those on normal ECs, we compared L1CAM expression in MDA-MB-231 cells and HUVECs. L1CAM expression was found to be higher in MDA-MB-231 cells than in HUVECs (Supplementary Fig. 10a, b). However, EC-L1CAM was highly suppressed under normal conditions and upregulated later (48 h after irradiation) (Supplementary Fig. 10a). In contrast to MDA-MB-231 cells, in HUVECs, we found that Myc expression was peaked at 1 h after irradiation during DNA damage checkpoint signalling, and then degraded. At 48 h, L1cam expression was significantly induced, and Myc expression was again increased in normal ECs (Supplementary Fig. 10b). These different L1CAM expression may contribute to the contrasting roles of L1CAM in cancer cells and normal ECs. From these results, we suggest that Ab417 not only can prevent DNA damage-induced cardiotoxicity without interfering with the anti-tumour effects, rather than enhancing anti-tumour effects.

**The anti-L1CAM antibody has therapeutic potential for DNA damage-induced cardiotoxicity.** Next, we aimed to evaluate if Ab417 could be used as a therapeutic treatment by assessing if it specifically accumulates in irradiated, damaged cardiac tissues but no in non-damaged cardiac tissues. To this end, mouse hearts were focally irradiated with an ablative high dose of 90 Gy using a 3-mm collimator. As expected, focally irradiated cardiac tissues showed extensive vessel damage, including inflammation, vascular destruction, and collagen accumulation; in addition, nuclear L1-CT was markedly observed in ECs of irradiated cardiac tissue (Supplementary Fig. 11a). To detect Ab417 accumulation, we radiolabelled Ab417 with $^{64}$Cu to obtain $^{64}$Cu-NOTA-Ab417[36] and intravenously injected $^{64}$Cu-NOTA-Ab417 into mice that were focally irradiated with 90 Gy. Positron emission tomography/computed tomography (PET/CT) scans revealed that $^{64}$Cu-NOTA-Ab417 accumulated specifically in irradiated hearts (Supplementary Fig. 11b). From these results, we suggest that Ab417 specifically accumulates in irradiation-damaged cardiac tissues via increased L1CAM expression.

To evaluate the clinical relevance of our findings, we analysed vascular L1CAM expression in cardiac tissues from patients with cardiomyopathy ($n = 12$) compared with that in normal cardiac tissues ($n = 8$). Although the patients did not receive radiotherapy

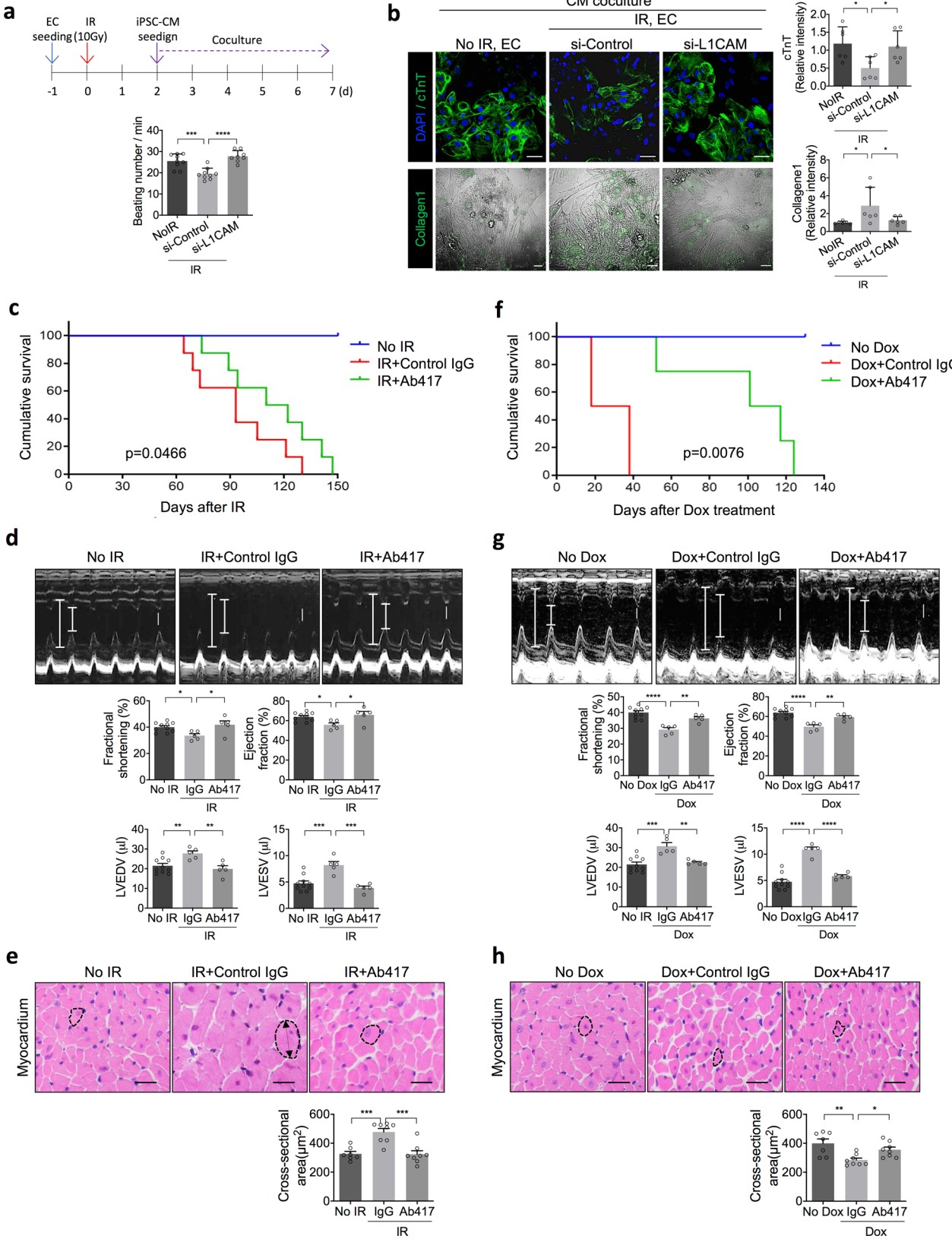

or Dox treatment, we selected tissues with vascular DNA damage foci ($n = 8$). Compared with tissues that did not exhibit DNA damage ($n = 4$) and normal cardiac tissues, patient tissues with increased γ-H2AX foci showed significantly higher levels of nuclear L1CAM foci in vascular ECs (Fig. 6a, b). Moreover, significantly

higher numbers of nuclear L1CAM foci were found in patient tissues with α-SMA⁺ CD31⁺ cells indicative of EndMT (Fig. 6c). These results confirm that L1-CT accumulates in the nuclei of vascular ECs of patients with cardiomyopathy, suggesting that Ab417 might be useful in these patients.

Fig. 5 An anti-L1CAM antibody (Ab417) mitigates irradiation (IR)- and Dox-induced cardiotoxicity and increases survival. a, b Co-culture of iPSC-CMs and irradiated ECs transfected with control or L1CAM siRNA. a Schematic of the procedure for co-culture of iPSC-CMs and irradiated ECs. ECs were transfected with control or L1CAM-specific siRNA and irradiated with 10 Gy. Forty-eight hours after IR, the ECs were co-cultured with iPSC-CMs (top). The beating rates of the iPSC-CMs were examined after 5 days of co-culture with irradiated ECs (bottom). Error bars represent mean ± SD (No IR vs. si-Control +IR $p = 0.0005$; si-Control+IR vs. si-L1CAM + IR $p > 0.0001$). b Immunofluorescence staining (left panel) and quantification (right panel) of cTnT expression in cardiomyocytes and the collagen I deposition area in co-cultured cells (magnification, ×200). Scale bar = 50 μm. Error bars represent mean ± SD (cTnT expression: No IR vs. si-Control+IR $p = 0.0239$; si-Control+IR vs. si-L1CAM + IR $p = 0.0475$, collagen I deposition: No IR vs. si-Control + IR $p = 0.027$; si-Control + IR vs. si-L1CAM + IR $p = 0492$). c–e Mice were injected intravenously with control IgG or Ab417 (10 mg/kg) three times a week for 2 weeks and received 16 Gy thoracic IR. f–h Mice were injected intravenously with control IgG or Ab417 (10 mg/kg) with or without intraperitoneal Dox injection (4 mg/kg) three times a week for 2 weeks. c, f Cumulative survival analysis measured in days after treatment ($n = 8$ animals per group). d, g Echocardiography results (upper panels) and quantification (lower panels) of FS (%), LVEF (%), LVESV (μL), and LVEDV (μL) (No IR $n = 10$; IR + IgG $n = 5$, IR + Ab417 $n = 5$; No Dox $n = 10$; Dox+IgG $n = 5$, Dox + Ab417 $n = 5$). Scale bar = 1 mm. Error bars represent mean ± SD (FS: No IR vs. IR + IgG $p = 0.0331$; IR + IgG vs. IR + Ab417 $p = 0.0172$; Dox+IgG vs. Dox + Ab417 $p = 0.0046$, LVEF: No IR vs. IR + IgG $p = 0.0284$; IR + IgG vs. IR + Ab417 $p = 0.0182$; Dox + IgG vs. Dox + Ab417 $p = 0.0027$, LVEDV: No IR vs. IR + IgG $p = 0.0099$; IR + IgG vs. IR + Ab417 $p = 0.0054$; No Dox vs. Dox $p = 0.0003$; Dox + IgG vs. Dox + Ab417 $p = 0.003$, LVESV: No IR vs. IR + IgG $p = 0.0005$; IR + IgG vs. IR + Ab417 $p = 0.0002$, ****$p < 0.0001$). e, h Haematoxylin and eosin−stained ventricular myocardium (upper panels) and quantification (lower panels) of cardiomyocyte cross-sectional area (No IR $n = 7$; IR + IgG $n = 8$; IR + Ab417 $n = 8$; No Dox $n = 7$; Dox + IgG $n = 8$; Dox+Ab417 $n = 8$) (magnification, ×400). Scale bar = 20 μm. Error bars represent mean ± SEM (No IR vs. IR + IgG $p = 0.0003$; IR + IgG vs. IR + Ab417 $p = 0.0002$; No Dox vs. Dos+IgG $p = 0.024$; Dox + IgG vs. Dox + Ab417 $p = 0.0493$). (a, d: log-rank Mantel-Cox test; all other panels: one-way ANOVA for multiple comparisons).

## Discussion

The common chemotherapy drug Dox and radiation therapy induces DNA damage and cardiotoxicity, and no therapy is currently available. In the present study, we found that nuclear L1-CT regulates persistent DNA damage in vascular ECs during the development of radiation- and Dox-induced cardiotoxicity and that anti-L1CAM antibody prevents DNA damage-induced cardiotoxicity. In vascular ECs, colocalization of L1-CT and γ-H2AX foci appeared to regulate persistent DNA damage. Nucleus L1-CT foci was also colocalized with DNA repair enzymes. Pre-treatment with an anti-L1CAM antibody significantly reduced nuclear L1-CT localisation in vascular ECs and the subsequent vascular damage caused by radiation and Dox treatment. Furthermore, nuclear colocalization of L1-CT and γ-H2AX foci was inhibited by the γ-secretase inhibitor L-685,458 and by introduction of a mutation in the NLS or RSLE motif of L1CAM. The results suggest that γ-secretase-mediated cleavage of L1CAM and nuclear translocation of L1-CT are required for colocalization of L1-CT and γ-H2AX foci, which results in vascular damage after radiation or Dox treatment. We are the first to suggest that L1CAM regulates DNA damage via colocalization with persistent γ-H2AX foci. The numbers of nuclear L1CAM foci were correlated with the total numbers of γ-H2AX foci.

In this study, we showed that the intensity and size of foci were increased after 24 h of irradiation. This finding might reflect the minor population of DNA damage from the irradiation, as well as the outcome of secondary celluar processes. The inability to repair DNA damage at telomeric regions is a major conrtibutor to a persistent DDR[54]. Moreover, DSB can also be formed indirectly from base excision repair (BER) of clustered lesions, DNA replication of SSB-containg templates or from transcription processes[55]. Therefore, various causes might contribute to the late occurrence of γ-H2AX foci.

The direct effects of Ab417 on L1CAM were investigated. In a previous report[36], Ab417 was found to decrease the membrane L1CAM levels via L1CAM endocytosis; in our study, we similarly showed that Ab417 internalisation induced the endocytotic lysosomal degradation of membrane full-length L1CAM, which reduced L1CAM expression and subsequently reduced the expression of the L1-CT fragment. Furthermore, in vivo, we found that the $^{64}$Cu-labelled anti-L1CAM antibody specifically marked focally irradiated heart regions, indicating that the anti-L1CAM antibody efficiently attenuates radiation-induced cardiac damage and can prevent cardiac damage. Dox and radiation

treatments each induced a significant deterioration of cardiac function parameters, including FS, LVEF, LVESV and LVEDV, which was prevented by treatment with the anti-L1CAM antibody. Notably, anti-L1CAM antibody treatment prolonged survival after radiation or Dox treatment by preventing cardiac dysfunction.

Ab417 regulates the L1CAM expression level that regulates malignant transition. L1CAM expression is tightly regulated in chemoresistant cells undergoing EMT[56]. HIF-1a and TGF-β1 are EMT regulators that also regulate L1CAM expression in cancer cells[57]. TGF-β1-induced L1CAM expression is related to apoptotic resistance and promotes the malignant transition of intestinal epithelial cells[58]. Our previous study showed that the radiation-induced EndMT process is mainly regulated by HIF-1a and TGF-β1 in normal ECs[38]. Thus, increased L1CAM expression after radiation may also be a part of natural mechanisms to survive with irreparable DNA damage, promoting EndMT. Ab417-treated cells exhibited reduced radiation-induced aneuploidy of normal cells, suggesting that Ab417 can regulate tumorigenic changes in DNA-damaged normal cells.

After radiation treatment, normal ECs appear as flattened cells with enlarged nuclei, exhibit a fibrotic phenotype and have a developed cytoskeleton[37,38]. As we have previously shown, these characteristics are accompanied by radiation-induced EndMT and associated tissue hypoxia[38]. In this study, we observed that persistent DNA damage in ECs associated with a fibrotic phenotype caused EndMT. Previous studies have supported this observation, as EndMT of cardiovascular cells has been reported to cause various types of cardiac dysfunction such as cardiac fibrosis and cardiomyopathy[59–61]. Indeed, we found that radiation- and Dox-induced EndMT may be correlated with tissue hypoxia and concurrent processes of cardiomyocyte damage, such as loss of TnT. In irradiated EC-p53KO mice, the numbers of L1-CT and DNA damage foci were markedly increased in ECs, consistent with EndMT and significant loss of cTnT in cardiomyocytes. Consistent with this possibility, our data indicate that EndMT of cardiovascular ECs may cause loss of function in iPSC-CMs co-cultured with irradiated ECs compared to non-irradiated ECs, leading to increased collagen I deposition on the extracellular matrix of co-cultured iPSC-CMs. Additionally, dysfunction of iPSC-CMs may be affected by co-culture with irradiated ECs showing increased collagen expression. Consistent with our findings, the extracellular matrix of cardiomyocytes has been reported to play a crucial role in cardiac homoeostasis[62–64], and

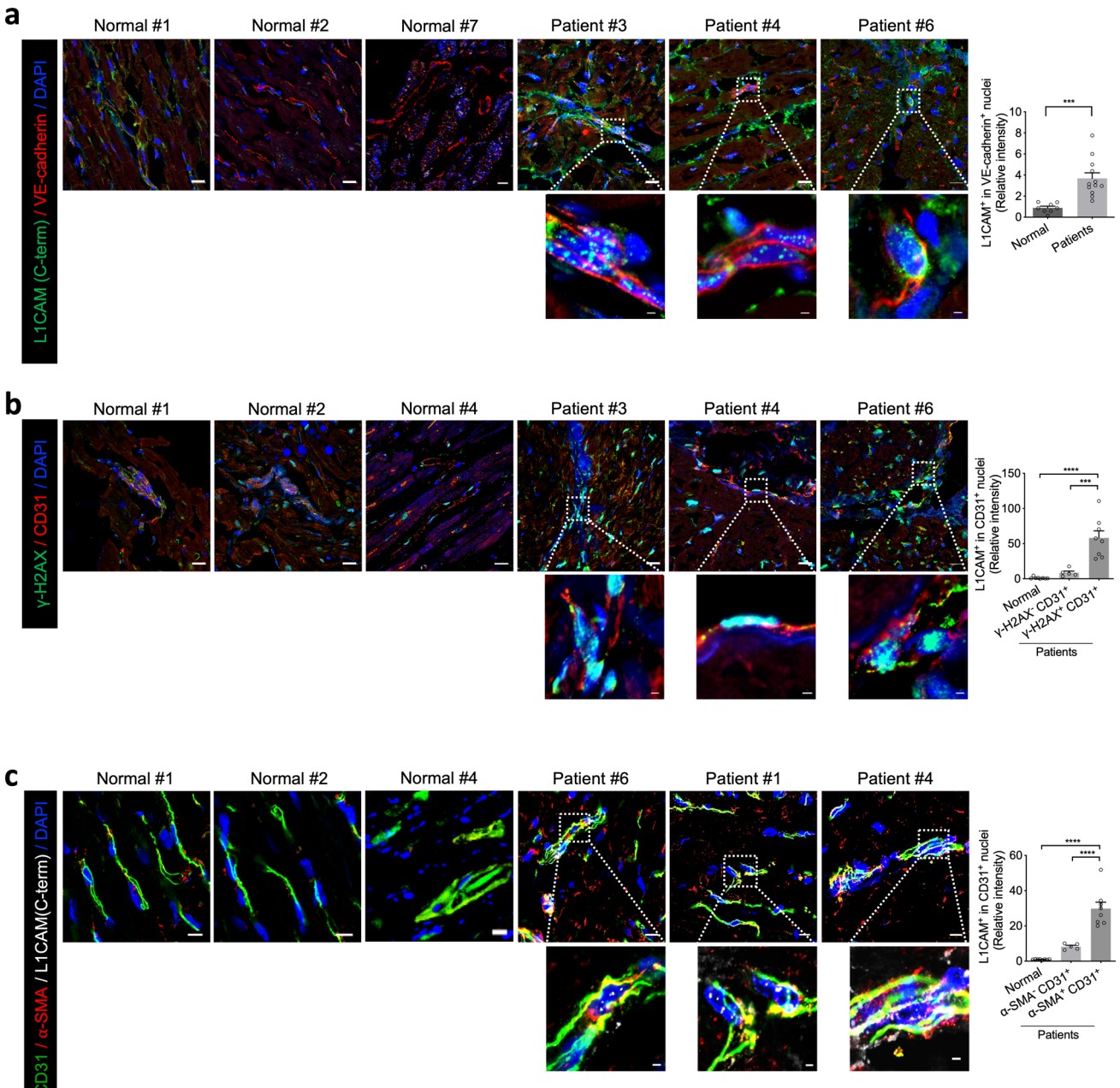

**Fig. 6 Nuclear L1CAM localisation with vascular EndMT and γ-H2AX L1CAM colocalization in heart tissues from patients with cardiomyopathy.**
**a** Immunofluorescence staining of L1CAM and VE-cadherin in heart tissues from patients with cardiomyopathy ($n = 12$) and patients without cardiomyopathy ($n = 8$) (magnification, ×400). Scale bar = 10 μm (enlarged, 2 μm). Error bars represent mean ± SEM ($p = 0.0004$).
**b** Immunofluorescence staining of γ-H2AX and CD31 in heart tissues from patients with cardiomyopathy and patients without cardiomyopathy (upper panels) and quantification (lower panel) of nuclear L1CAM$^+$ cells among CD31$^+$ cells (magnification, ×400). Scale bar = 10 μm (enlarged, 2 μm). Error bars represent mean ± SEM (Normal vs. γ-H2AX$^+$ CD31$^+$ $p > 0.0001$; γ-H2AX$^-$ CD31$^+$ vs. γ-H2AX$^+$ CD31$^+$ $p = 0.0003$) **c** Immunofluorescence staining of L1CAM, α-SMA, and CD31 in heart tissues from patients with cardiomyopathy and from patients without cardiomyopathy (upper panels) and quantification (lower panel) of nuclear L1CAM$^+$ cells among α-SMA$^-$CD31$^+$ and α-SMA$^+$CD31$^+$ cells (magnification, ×400). Scale bar = 10 μm (enlarged, 2 μm). Error bars represent mean ± SEM ($p > 0.0001$). (**a** two-tailed student's $t$-test; **b**, **c** one-way ANOVA for multiple comparisons).

the imbalance of collagen deposition in the cardiac interstitium may cause systolic dysfunction[63]. Furthermore, an increase in collagen deposition has been reported to cause cardiomyocyte dysfunction[64].

Next, Ab417 targeting of L1CAM in ECs reduced DNA damage, vascular EndMT, hypoxia, and cTNT loss in vivo after Dox treatment or radiation. Thus, we theorise that DNA damage in ECs may cause vascular dysfunction arising from EndMT, subsequently increasing hypoxia and reducing the

cTnT level in cardiomyocytes, ultimately affecting overall heart function.

Additionally, we focused on the effects of cardiac angiocrine factors, such as NRG-1, on cardiomyocyte protection. It has been reported that cardiac angiocrines, secretory factors of ECs, mediate EC-CM interactions by paracrine signalling[65]. Of these cardiac angiocrines, NRG-1 is known to protect cardiomyocytes from ischaemic injury and chemotherapeutic agents[52,53]. Secretory factors from normal ECs, but not those from irradiated ECs

undergoing EndMT, can reduce cardiomyocyte damage. In particular, our findings suggest that the angiocrine NRG-1 can protect cardiomyocytes from irradiation.

Furthermore, we investigated whether the L1-CT and DNA damage responses associated with cardiac damage have clinical relevance by utilising tissues from patients with other cardiac diseases. Our data revealed that cardiovascular ECs with persistent DNA damage showed upregulated L1CAM expression with increased EndMT occurrence, which can be attributed to the development of cardiomyopathy. Indeed, cardiac EndMT has been reported to affect cardiac fibrosis, cardiovascular disease, and cardiomyopathy[59,60,66]. Therefore, these findings indicate that the anti-L1CAM mAb may be a potential therapeutic agent for regulation of cardiomyopathy.

In addition to the potential use of this antibody for cardiomyopathy, we recently demonstrated that combining Ab417 treatment with gemcitabine or cisplatin treatment enhances tumour growth inhibition in cholangiocarcinoma[36]. Moreover, we have reported the antitumor effects of a murine anti-L1CAM mAb on intrahepatic cholangiocarcinoma[67] and the feasibility of radioimmunotherapy using radiolabelled anti-L1CAM antibody in cholangiocarcinoma[68]. Our data also showed that Ab417 markedly enhanced the anti-tumour effects of radiation and Dox treatment, indicating that Ab417 can prevent cardiotoxicity by enhancing the efficacy of anticancer therapy. However, the differential function of L1CAM in tumour and normal cells remains undefined. It seems that cancer cells including MDA-MB-231 cells exhibit aberrant expression of L1CAM. Furthermore, in glioblastoma multiforme (GBM) cancer cells, highly expressed L1CAM was previously found to increase DNA damage checkpoint activation by NBS1 upregulation via c-MYC 3 h after radiation, resulting in radioresistance[33]. However, in ECs, L1CAM and c-Myc expression was upregulated after induction, and L1CAM and r-H2AX colocalized in the late stage (48 h after radiation) in our study, indicating persistent DNA damage. According to recent reports, Myc influences genomic instability in normal cells[69,70]. Therefore, we cautiously hypothesise that the opposite responses of L1CAM in tumour cells and normal ECs may be due to the dual nature of Myc (i.e., checkpoint signalling activation and survival in cancer cells vs. genomic instability in normal cells). However, the nature of these differential mechanisms of L1CAM and the precise interactions between L1-CT and colocalized DNA damage repair proteins remain unclear, and we are currently investigating these topics in another study. Nevertheless, we have shown that regulation of vascular DNA damage via targeting of endothelial L1CAM can attenuate radiation- and Dox-induced cardiotoxicity.

Collectively, our findings support L1CAM-targeted therapy as a potential strategy to overcome the irreparable and persistent DNA damage that occurs during the development of cardiotoxicity from radiation or chemotherapeutic agents, including Dox. In particular, the use of an L1CAM-targeted antibody that prevents the nuclear translocation of L1CAM may be a novel strategy to prevent cardiotoxicity during DNA damage-based anticancer therapy. Further studies on the precise molecular mechanisms underlying the role of vascular L1CAM in DNA damage will aid in the development of novel therapeutic strategies for cardiac diseases.

## Methods

**Mice and ethical approval**. All animal experiments were approved by the Institutional Animal Care and Use Committee of the Korea Institute of Radiological & Medical Sciences (Kirams 2016-0039, 2017-0007) and are reported in accordance with the Animal Research: Reporting of In Vivo Experiments (ARRIVE) guidelines.

Specific pathogen-free female BALB/c nude mice were purchased from Orient Bio. Specific pathogen-free C57BL/6 Tie2-Cre and Trp53[flox/flox] mice were purchased from the Jackson Laboratory. Male Tie2-Cre and Tie2-Cre;Trp53[flox/+] mice were crossed with female Trp53[flox/flox] mice to generate Tie2-Cre;Trp53[flox/+] and Tie2-Cre;Trp53[flox/flox] mice, respectively. Tie2-Cre;Trp53[+/+] or Tie2-Cre[−/−] littermates were used as controls. All animal experimental data provided are representative of three independent experiments. All experiments were conducted with 6–8-week-old mice, and the mice were maintained on a 12 h light-dark cycle in a standard environment (20 ± 1 °C room temperature, 50 ± 10 % relative humidity) with a standard diet and water ad libitum. All mice were anaesthetised with a combination of anaesthetics before being euthanized.

**Human tissue specimens**. Cardiac tissues ($n = 14$) were purchased from OriGene. Six patients tissues of Normal #3-Normal #8 were obtained from Heart tissue array (US Biomax, BC30013). Eight patients (five male and three female) showed normal clinicopathological characteristics, and twelve exhibited cardiomyopathy (four males and eight females; Supplementary Table 1).

**Production of Ab417**. The antibody Ab4 was selected from a human naive Fab library, and Ab4M was generated by increasing the affinity of Ab4 45-fold via mutation of three residues in the complementarity-determining regions (CDRs). Next, the human anti-L1CAM mAb Ab417 was generated by increasing the affinity of Ab4M via yeast display of scFv containing randomly mutated light chain CDR3. Ab417 was found to be cross-reactive with mouse L1CAM[35].

The anti-L1CAM antibody Ab417[35] was purified from the culture supernatants of transfected CHO cells. Cells expressing Ab417 were cultured in MEM-α (WELGENE) supplemented with 5% (v/v) dialysed FBS (Thermo Fisher Scientific) under 5% $CO_2$, and the medium was changed to serum-free medium (SFM4CHO, GE Life Sciences) for antibody production. The culture supernatant was centrifuged and filtered using a Sartolab bottle-top filter (0.22 μm, PES; Sartorius) and then subjected to affinity chromatography on a protein A-agarose column (GenScript) for purification. The protein A-bound antibodies were eluted using 0.1 M sodium citrate containing 150 mM sodium chloride (pH 3.6). The eluted antibodies were stored after a buffer change to 25 mM sodium citrate containing 150 mM sodium chloride and 0.007% Tween 20 (pH 6.4). The protein concentrations were determined using a NanoDrop 2000 UV-Vis Spectrophotometer (Thermo Fisher Scientific).

**Mouse experiments**. Mice were randomly assigned to each group. To establish a Dox-induced cardiotoxicity model, mice were intraperitoneally injected with Dox (4 mg/kg, 6 times/2 weeks); controls were injected with the vehicle. To establish a radiation-induced cardiotoxicity model, the thoracic cavities of mice were irradiated with 16 Gy (BALB/c nude mice) using the X-RAD 320 platform (Precision X-ray). Ab417 (10 mg/kg, 6 times/2 weeks) was injected intravenously into BALB/c nude mice 1 h before the start of Dox or radiation treatment. Control mice were administered an equivalent dose of an isotype control antibody (IgG). The serum concentrations of CRP (catalogue no. MCRP00; R&D Systems), E-selectin, and ICAM-1 (catalogue no. ab171182 and ab100688; Abcam) were measured using mouse Quantikine ELISA kits.

**Echocardiography**. Cardiac function was assessed using a high-resolution echocardiography system (Vivid 7; GE Medical Systems). Mice were anaesthetised using a mixture of Zoletil (30 mg/kg) and Rompun (10 mg/kg) via intraperitoneal injection. The percentage of LV fractional shortening (FS) was calculated using M-mode. The LVEF percentage was calculated as [(diastolic volume − systolic volume)/diastolic volume] × 100. Three different cardiac cycles were measured for each assessment.

**Histology and immunohistochemistry**. Tissues were fixed in 10% (v/v) neutral-buffered formalin, embedded in paraffin, and sectioned. The sections were deparaffinized and stained as previously described[38]. Collagen deposition was assessed using Masson's trichrome stain (Sigma-Aldrich). Haematoxylin and eosin–stained slides were analysed to score the grade of ventricular inflammation. Histological scores were determined as follows: grade 0, no inflammation; grade 1, <10% of the heart section infiltrated; grade 2, 10–30%; grade 3, 30–50%; grade 4, 50–90%; and grade 5, >90%. At least five images per section were acquired for quantification, and the positively stained areas were evaluated using ImageJ software 1.49 v. (http://imagej.net/).

**Antibodies for immunoblotting and immunohistochemistry**. Immunoblotting, immunohistochemistry, and immunofluorescence staining were performed using primary antibodies against the N-terminal domain of L1CAM (immunohistochemistry/immunofluorescence 1:100; sc-31032; Santa Cruz Biotechnology), L1-CT (immunohistochemistry/immunofluorescence 1:500, immunoblotting 1:1000; LS-B9803; LSBio; immunofluorescence 1:200; sc-53386; Santa Cruz Biotechnology; immunofluorescence 1:200; ab123990; Abcam), complete L1CAM (immunoblotting 1:1000; sc-53386; Santa Cruz Biotechnology), γ-H2AX (immunohistochemistry/immunofluorescence 1:500, immunoblotting 1:1000; 05-636; Millipore), CD31

(immunohistochemistry/immunofluorescence 1:200; #AF3628; R&D Systems; immunoblotting 1:1000; 28364; Abcam), GAPDH (immunoblotting 1:1000; sc-47724; Santa Cruz Biotechnology), lamin B (immunoblotting 1:1000; sc-6216; Santa Cruz Biotechnology), αSMA (1:2000; A5228; Sigma-Aldrich), VCAM-1 (immunoblotting 1:1000; sc-1504; Santa Cruz Biotechnology), p-ATM (immuno-fluorescence 1:200; 05-740; Millipore), 53BP1 (immunofluorescence 1:200; sc-10914; Santa Cruz Biotechnology), and DNA-PKcs (immunofluorescence 1:200; sc-135886; Santa Cruz Biotechnology). To visualise actin stress fibres, cells were stained with Alexa Fluor 488-conjugated phalloidin (1:40; Invitrogen), which specifically binds polymerised F-actin. At least five images per section were acquired for quantification, and the positively stained areas were evaluated using ImageJ.

**Cell culture and treatments**. HUVECs (#C-12203) and human CD14 + mono-cytes (#C-12909) were obtained from PromoCell and cultured in Endothelial Cell Growth Medium 2 and Mononuclear Cell Medium (PromoCell), respectively, under 5% $CO_2$. For silencing experiments, cells were transfected with siRNAs targeting TP53 and L1CAM, as well as control siRNA (Santa Cruz Biotechnology) using Lipofectamine 2000 (Invitrogen) according to the manufacturer's instruc-tions. Cells were irradiated with γ-rays from a 137Cs source (Atomic Energy of Canada) at 3.81 Gy/min or treated with 0.5 μM Dox. Before irradiation or Dox treatment, the cells were pre-treated with the anti-L1CAM antibody Ab417 (20 μg/mL) for 1 h or treated with 3 μM of the γ-secretase inhibitor L-685,458 (sc-204042; Santa Cruz Biotechnology). For analysis of cellular proteins, immunocytochemistry was performed as previously described[38]. Cytoplasmic and nuclear proteins were extracted using NE-PER™ Nuclear and Cytoplasmic Extraction Reagents (#78833; Thermo Fisher Scientific).

**RT-qPCR analysis**. RNA was isolated using TRI reagent (MRC), after which 1 μg of RNA was used to synthesise cDNA with an Omniscript RT Kit (Qiagen). PCR was conducted in triplicate using a CFX96™ Real-Time PCR Detection System (Bio-Rad Laboratories) and qPCR SYBR Green Master Mix (Invitrogen). For each sample, the target gene expression was normalised against the geometric mean of the expression of the reference gene GAPDH. The following primers were used: L1CAM (NM_000425) forward, 5′- GGTTCGTTCATTGGCCAGTACAGT -3′, and reverse, 5′- GTCAGGGAGCAAGAAAGACAGCAT -3′; and GAPDH (NM_002046) forward, 5′-CGAGATCCCTCCAAAATCAA-3′, and reverse, 5′-CCTTCTCCATGGTGGTGAA-3′.

**RNA-seq analysis**. Total RNA was isolated from HUVECs, and RNA quality was assessed using an Agilent 2100 Bioanalyzer (Agilent Technologies). RNA-seq libraries were constructed using an NEBNext Ultra II Directional RNA-Seq Kit (New England Biolabs, Inc., UK) according to the manufacturer's instructions and were sequenced in 100 bp paired-end runs using a HiSeq X10 platform (Illumina, Inc., USA). The RNA-seq reads were mapped to University of California, Santa Cruz (UCSC) genome hp19 using TopHat software. Genes with a maximal log2-transformed read count <3 across samples were excluded from the analysis. We used the criterion of a | log2(fold change, FC) | > 1 to identify the differentially expressed genes by comparing the irradiated group to the non-irradiated group. EndMT-related genes among the differentially expressed genes were displayed in R using the pheatmap package (v1.0.12). The RNA-seq data generated in this study are available in the Gene Expression Omnibus (GEO, accession number GSE155285).

**HR/NHEJ assay**. The plasmids pHPRT-DR-GFP, pimEJ5GFP, and pBAD-I-SceI (#26476, #44026, and #60960, respectively; Addgene)[46–48] were kindly donated by Dr. Chang-Woo Lee (Department of Molecular Cell Biology, Sungkyunkwan University School of Medicine). pHPRT-DR-GFP or pimEJ5GFP was transfected into HUVECs along with control or L1CAM-targeting siRNA using Lipofectamine 2000 reagent (Invitrogen). After 1 day, the cells were transfected with pBAD-I-SceI to induce DNA damage and cultured for 48 h. The cells positive for GFP because of DNA repair were analysed on a FACSCalibur flow cytometer (Becton Dickinson, Franklin Lakes, NJ) to estimate the frequencies of HR and NHEJ. Gating strategy for flow cytometry analysis was shown in Supplementary Figure 12.

**Co-culture of iPSC-CMs and ECs**. iPSC-CMs were purchased from Cellartis (#Y50025) and cultured in 50 μg/mL fibronectin-coated dishes according to the manufacturer's protocols. HUVECs were transfected with control or L1CAM-targeting siRNA using Lipofectamine 2000 reagent (Invitrogen). Forty-eight hours after seeding ($5 × 10^3$ cells), irradiated HUVECs or non-irradiated HUVECs were co-cultured with iPSC-CMs for 5 days in 35 mm confocal dishes. The cell seeding density of the iPSC-CMs (alone and in co-cultures) was $4 × 10^5$ cells/dish (1:80 ratio for iPSC-CMs:HUVECs). The ECs were labelled with the tracking dye PKH26 (Sigma # PKH26GL). The cells were cultured at 37 under 5% $CO_2$ in culture medium (iPSC-CMs: MiraCell® CM Culture Medium v2). The medium was changed after 2 days. Images of beating iPSC-CMs were obtained with a micro-scope and were measured for 1 min. Cells were fixed in 10% (v/v) neutral-buffered formalin and subjected to TnT and collagen staining for immunofluorescence analysis.

**Statistical analysis**. The data are expressed as the means ± SDs or SEMs. A log-rank test was used for the survival assay, unpaired Student's t-test was used to analyse differences between two groups, and one-way ANOVA followed by Tukey's multiple comparisons test was used for multiple comparisons (>2 groups). p-values < 0.05 were considered to indicate statistical significance. Statistical analysis was performed using GraphPad Prism version 7.0. The experimenters were blinded to the group assignments and outcome assessments. We used Image J 1.49 v (NIH), R 4.0.4, Zen 3.2 (Zeiss) and Diva 7.0 (BD Biosciences) for data and image analysis and plotting.

Additional information is available in the Supplementary Methods.

**Reporting summary**. Further information on research design is available in the Nature Research Reporting Summary linked to this article.

## Data availability
The RNA-seq data was deposited in Gene Expression Omnibus (GEO) with accession number "GSE155285". All other relevant data are available from the corresponding author upon reasonable request. Source data are provided with this paper.

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

## Acknowledgements

This work was supported by grants from the National Research Foundation (NRF-2017M2A2A7A02019482, NRF-2020M2D9A2093964, NRF-2020M2C8A2069337, NRF-2018R1D1A1A09084274 and NRF-2020R1A2B5B02002709) and the Korea Institute of Radiological & Medical Sciences (KIRAMS, 50531-2021) funded by the Ministry of Science and ICT (MSIT), Republic of Korea.

## Author contributions

Y.J.L., H.J.H., and S.H.C. conceived and designed the study. Y.J.L., S.H.C., A.R.K., J.K.N., S.C., and T.S.L. performed the experiments with help from J.H.K., K.J.C., J.W.L., H.J.C., Y.W.K., H.J.L., K.S.K., and J.H.C. Y.J.L., S.H.C., A.R.K., S.B., J.K., and H.J.H. interpreted the data. Y.J.L., H.J.H., and S.H.C. drafted the manuscript.

## Competing interests

The authors declare no competing interests.
