## [Peer Review File · Nature Communications]

Reviewers' Comments:

Reviewer #1:

Remarks to the Author:

This paper reports that a monoclonal antibody against L1 cell adhesion molecule (L1CAM) (a) reduces radiation- and doxorubicin-induced EMT and DNA damage in endothelial cells; and that (b) reduction of endothelial cell DNA damage lessens cardiomyocytes dysfunction and heart failure from radiation and doxorubicin.

The topic of protecting the heart against cancer therapies, including doxorubicin, is important. The major conclusions of this paper are novel.

That said, the mechanistic dissection in this paper is quite poor. There are holes in many key steps. In addition, the effects of L1CAM inhibition on some of these steps is directionally opposite in endothelial cells as in other cell types. While the latter is certainly possible, the data provide no insight into why this is the case. These mechanistic deficiencies markedly decrease enthusiasm for this paper.

Specific points

1. It is not at all clear how the L1CAM Ab brings about its reported effects. Previous work by this group reports that the antibody is internalized and works from within the cell. But, what it actually does is not well defined here or in their previous work. The authors state in the discussion that that the antibody inhibits gamma-secretase-mediated cleavage L1CAM, an event that is needed to release the C-terminal fragment to the nucleus where it localizes with foci of DNA damage. But, I do not see the data supporting this in the paper. In fact, the data presented suggest that levels of both full-length L1CAM and the C-terminal fragment are decreased in antibody-treated cells – suggesting that internalization of L1CAM destabilizes both full length and the C-terminal fragment. Thus, while other data in the current paper shows that interfering with cleavage and nuclear translocation of L1CAM phenocopies what the antibody, the authors do not precisely define what the antibody actually does.

2. Disruption of L1CAM signaling to the nucleus appears to impact DNA damage in directionally opposite ways in endothelial and cancer cells (ref 30). Specifically, the L1CAM antibody decreases DNA damage in endothelial cells perhaps explained by increase DNA repair. In contrast, loss of L1CAM signaling to the nucleus increases DNA damage in the cancer cells (ref 30 in the text) thereby rendering the cancer cells more susceptible to killing. While it is possible to have different effects in non-transformed versus cancer cells, the basis is not investigated or discussed. Moreover, early markers of activation of DNA damage response are decreased in both the endothelial cell (this paper) and cancer cell (ref 30) contexts. None of these issues are sorted out.

3. The next hole in mechanism concerns how endothelial cell DNA damage brings about cardiomyocyte dysfunction. To the authors' credit, they undertook co-culture experiments with stressed endothelial cells and human iPS cell-derived cardiomyocytes. The authors posit that endothelial DNA damage impacts cardiomyocytes through changes in extracellular matrix and hypoxia. But the data presented are insufficient in my mind to define any mechanism. There are also some related concerns. First, iPS cell-derived cardiomyocytes are very immature cells and may not be the correct cells with which to be asking this question in. Second, the loss of troponin from the cardiomyocytes, which the authors interpret as cardiomyocytes dysfunction, may well reflect cardiomyocyte death. Third, the data supporting that the effects of radiation are limited to endothelial cells – not cardiomyocytes – are not so clear.

4. Specific comments on the data:

- a. Fig.1c label on graph says 72h whereas main text says 48h.
- b. Fig.1d if L1CAM colocalization with γ -H2AX foci orchestrates persistent DNA damage, why DNA DSB goes down in 24h post IR while number of L1CAM foci goes up?
- c. Fig.1f is the quantification done on the cytosolic, nuclear or total level?
- d. Phalloidin stains F-actin. Why are you using it as a marker of fibrosis?
- e. Fig.1g and 1j, should also plot individually number of γ -H2AX foci and number of L1CAM foci.
- f. Supp Fig.1d surprising how Ab417 is able to limit DNA damage but has no effect on cell death.

Should test more time points and also test in doxorubicin model.

g. Supp Fig.1e legend duplication error.

h. Fig. 5. FS (one dimension) and EF (3 dimensions) are essentially the same parameter. Need to also specify LVESV and LVEDV.

5. Not completely accurate that no therapeutic targets have been identified in doxorubicin-associated cardiomyopathy. See *Sci Transl Med*, 2019. 11 (478): eaau8866. PMID: 30728290. And *Nat Cancer*, 2020. 1(3): 315-328. PMID: 32776015.

Reviewer #2:

Remarks to the Author:

The authors applied the anti-L1CAM antibody (Ab417, affinity matured Ab4) to attenuate cardiotoxicity after treatment by thoracic irradiation and doxorubicin. They found that vascular endothelial cells showed fibrotic phenotype cells after treatment and that L1-CAM was colocalized with DNA damage foci and it was involved in persistent DNA damage. Their strategy is that administering an anti L1CAM antibody can reduce the overall expression level of L1-CAM on the surface of cardiovascular or vascular ECs and it may lead to a decrease in the number of DNA damage foci.

Comments

1. The authors suggested that Ab417 may inhibit endocytosis of L1CAM, but they should clarify where the Ab418-L1CAM complex is. While the complex remains on the surface of ECs, IgG1 isotype of Ab418 can lead to effector functions like ADCC, CDC, and ADCP, resulting in severe damage to ECs by immune cells and disrupting vasculature. Perhaps, authors need to conduct a mouse study with IgG4 isotype of Ab418 or LALA-PG mutants of IgG1 Fc to compare. In contrast, if the complex was not removed, it will be recycled by FcRn because the endothelial cells normally express FcRn to keep long-serum half-life of antibodies. So, the L1CAM expression level can be restored quickly unless Ab418 binding is pH-dependent. Need to address the possibility for those unwanted effects mediated by the anti-L1CAM antibody.

2. I would suggest discussing which mechanism enhanced L1CAM expression after irradiation and Dox treatment. The increased expression level of L1CAM can be also part of natural mechanisms to prevent cell growth and induce cell death as DNA-damaged cells can transform to tumor cells. In this context, did the EC or other cells treated by the anti-L1CAM antibody have normal DNA after repairing or still have damaged DNA? I am concerned that L1CAM antibody treatment can lead to tumor generation.

3. The author may need to demonstrate actual benefits in tumor-bearing mouse models, not in normal mice. I wonder whether the Dox and irradiation enhance the expression level of L1CAM on tumor cells because the L1CAM antibody treatment also can give a benefit to tumor cells for surviving through the same mechanism as for the ECs.

Reviewer #3:

Remarks to the Author:

The paper by Nam and colleagues show that vascular endothelial cells after irradiation or doxorubicin show persistent DNA damage foci, co-localizing with C-terminal domain of L1CAM. Additionally, they show that treatment with anti-L1CAM antibody decreased L1CAM overexpression and nuclear translocation in VECs, as well as reduced the number of DNA damage foci. After whole-heart irradiation, VEC-specific p53 deletion increased L1CAM/DNA damage foci colocalization and vascular fibrosis, but these effects were attenuated by Ab417 treatment- this also led to improvements in cardiac function and prolonged survival.

Overall, It is a potentially important study (which provides both some mechanistic insight as well as therapeutic opportunity) and the authors have used multiple approaches to support their conclusions (using both in vitro and in vivo models). Mechanistically, I think that there are several gaps which still need to be addressed- particularly the relationship between DDR and vascular

fibrosis and the detailed mechanisms by which L1CAM impacts on DNA repair. However, I think no single paper can address all questions. Another point is the apparent proposal that effects of irradiation are specific to vascular cells in the heart (it would be difficult to understand why other cell-types in the heart would not be able to mount a DDR after such as high dose of irradiation). There are some aspects which I believe need to be addressed so that publication can be supported.

Figure 1- The authors do not show convincing evidence that the DDR foci are persistent. All data relies on quantification of DDR foci in fixed cells at different time points. In order to make a point about a persistent DNA damage response – DDR reporters and live-cell imaging needs to be performed allowing the tracking of the lifespan of individual foci. Additionally, longer time points should be analyzed- see for instance previous reports have indicated that DDR foci can be persistent even months following irradiation (PMID: 22426077; PMID: 22426229). The kinetic window provided is too narrow.

- Also, I am wondering how the authors explain that the DDR decreased at 24h and then increased again at 48h after 10Gy irradiation (this would, in my view, indicate that these are newly generated DDR foci – possibly the outcome of secondary processes and not the ones generated as a result of the radiation).

- KD efficiency of L1CAM by siRNA is very mild (western blot not particularly convincing in my view). Have the authors considered other potentially more effective (non transient) ways to KD L1CAM (for instance using lentiviral mediated shRNA or CRISPR/CAS9). I am also not entirely sure what was the point of showing the siRNA experiments- if no other outcomes were shown in the main Figure.

(minor points) In the text authors indicate that RNA-seq showed increased expression of EndMT-related genes at 48h- however, in the Figure 1c 72h is indicated.

Co-localization of L1CAM and γH2A.X is not evident in small Figure panels- I would advise showing amplified co-localized foci and indicating co-localization with arrows.

Quality of Immunofluorescence in dox treatment is not particularly good- no clear large L1CAM foci are observed- why is that?

Could other confirmatory methods such as Immunoprecipitation be performed to further confirm the association between γH2A.X and L1CAM?

Effects of Ab417 on phalloidin expression are relatively mild- are there effects of Ab417 cell on size (it seems so from pictures)? What about cell proliferation- since DDR is affected it is reasonable to assume that cells may overcome cell-cycle arrest induced by DNA damage.

Figure 2- b) Western showing L1CAM expression following transfection- loading control is necessary. Also, the necessary controls should be added? Authors mentioned they have used lentivirus- mediated shRNA against L1CAM and then reintroduced L1CAM WT or mutant forms- however, there is no evidence of KD efficiency. Cells expressing a shRNA control should be included in the same western blot.

Immunofluorescence images for pATM and 53BP1- authors should better indicate co-localize foci (possibly showing magnifications). Why is p-ATM cytosolic in no IR cells (Supplementary Figure 2)?

Quality of DNA-PKcs immunofluorescence is not ideal (I am not sure how the authors were capable of detecting/quantifying individual foci). Other methods for determining co-localization should be employed here (Pearson's correlation coefficient?)

What do the authors mean with (Line247) persistent DNA damage repair?

Typo (line 197)

Figure 3- Authors show that knock-down of p53 in HUVECs show increased L1CAM expression and increased co-localization with γH2A.X after IR. I am wondering if p53KD impacts on cell death and proliferation following IR. (this point is also relevant for the in vivo experiments in Figure 3 and 4) d- immunofluorescent images for L1CAM by IF in hearts from wt and p53KO mice are not very convincing. Is there increased L1CAM/γH2A.X co-localization?

Supplementary Figure 3- Authors claim that DDR is not present in cardiomyocytes. Quality of the γH2A.X and WGA staining shown is not particularly convincing. Besides this is at odds with other reports. Activation of a DDR in cardiomyocytes has been shown to be a mechanism contributing to their post-mitotic state during differentiation PMID: 24766806. γH2A.X foci did not change with age and were detected in almost all adult CMs irrespective of age PMID: 30737259. Thus, even if γH2A.X was not induced in cardiomyocytes following irradiation (which is surprising and not sure why it would happen)- it should be present in adult cardiomyocytes irrespectively. I wonder if this is a technical issue.

Figure 6- can statistics be performed with $n=2$ controls?

Responses to the Reviewers' comments

We thank the reviewers for the helpful comments and suggestions, which have greatly helped us in improving the manuscript. Please find our point-by-point responses below. Revisions in the manuscript are highlighted in blue.

Reviewer #1 (Remarks to the Author):

This paper reports that a monoclonal antibody against LI cell adhesion molecule (LICAM) (a) reduces radiation- and doxorubicin-induced EMT and DNA damage in endothelial cells; and that (b) reduction of endothelial cell DNA damage lessens cardiomyocytes dysfunction and heart failure from radiation and doxorubicin. The topic of protecting the heart against cancer therapies, including doxorubicin, is important. The major conclusions of this paper are novel. That said, the mechanistic dissection in this paper is quite poor. There are holes in many key steps. In addition, the effects of LICAM inhibition on some of these steps is directionally opposite in endothelial cells as in other cell types. While the latter is certainly possible, the data provide no insight into why this is the case. These mechanistic deficiencies markedly decrease enthusiasm for this paper.

Specific points

1) It is not at all clear how the LICAM Ab brings about its reported effects. Previous work by this group reports that the antibody is internalized and works from within the cell. But, what it actually does is not well defined here or in their previous work. The authors state in the discussion that that the antibody inhibits gamma-secretase-mediated cleavage LICAM, an event that is needed to release the C-terminal fragment to the nucleus where it localizes with foci of DNA damage. But, I do not see the data supporting this in the paper. In fact, the data presented suggest that levels of both full-length LICAM and the C-terminal fragment are decreased in antibody-treated cells – suggesting that internalization of LICAM destabilizes both full length and the C-terminal fragment. Thus, while other data in the current paper shows that interfering with cleavage and nuclear translocation of LICAM phenocopies what the antibody, the authors do not precisely define what the antibody actually does.

Response: We apologize for the confusing representation of the results. In this study, we showed that gamma-secretase mediated the cleavage of LICAM in Fig. 2, but we did not show whether the Ab417 antibody inhibits the gamma-secretase-mediated cleavage of LICAM. To address how Ab417 directly affects LICAM, we have provided additional data of the effects of Ab417 on the endocytosis and lysosomal degradation of LICAM in Supplementary Fig. 4a-c.

The text in the Results section (p. 11) has been added as follows ;

Ab417 promotes the endocytic internalization and lysosomal degradation of LICAM

In accordance with our previous work³⁶, Ab417 inhibited the expression levels of full-length L1CAM and its C-terminal fragment in this study. L1CAM has also been reported to be regulated by its lysosomal degradation and endocytosis⁴⁹. To clarify how Ab417 directly affects L1CAM levels, we examined the effects of Ab417 on the endocytosis and lysosomal degradation of L1CAM. Ab417 was labelled with pHrodo fluorescence dye, which increases fluorescence upon internalization into acidic endosomes and lysosomes. The red fluorescence of internalized pHrodo-labelled Ab417 was observed at 1 h and peaked at 18 h after incubation in HUVECs (Supplementary Fig. 4a). Additionally, we examined whether Ab417 can directly induce the endocytotic internalization of endogenous L1CAM, resulting in lysosomal degradation. An anti-L1CAM antibody against endogenous L1CAM was labelled with green pHrodo dye. The pHrodo Green-labelled anti-L1CAM antibody colocalized with endocytic vesicles containing Ab417 with red fluorescence (Supplementary Fig. 4a), suggesting that Ab417 induces the internalization and endocytosis of membrane L1CAM. Ab417 significantly reduced L1CAM expression in WT-L1CAM-overexpressing HUVECs, and this effect was inhibited by treatment with the lysosomal degradation inhibitor bafilomycin-A1 (Supplementary Fig. 4b and c). These results suggest that Ab417 promotes the internalization, endocytosis and subsequent lysosomal degradation of L1CAM in HUVECs.

Together, our findings suggest that Ab417 reduced L1CAM expression by the endocytotic lysosomal degradation of full-length L1CAM and subsequently reduced the expression of the L1-CT fragment. Therefore, the Ab417 antibody inhibits persistent DNA damage by reducing the expression of full-length L1CAM and the subsequent nuclear L-CT fragment and does not directly interfere with the cleavage and nuclear translocation of L1CAM.

2) Disruption of L1CAM signaling to the nucleus appears to impact DNA damage in directionally opposite ways in endothelial and cancer cells (ref 30). Specifically, the L1CAM antibody decreases DNA damage in endothelial cells perhaps explained by increase DNA repair. In contrast, loss of L1CAM signaling to the nucleus increases DNA damage in the cancer cells (ref 30 in the text) thereby rendering the cancer cells more susceptible to killing. While it is possible to have different effects in non-transformed versus cancer cells, the basis is not investigated or discussed. Moreover, early markers of activation of DNA damage response are decreased in both the endothelial cell (this paper) and cancer cell (ref 30) contexts. None of these issues are sorted out.

Response: We thank the reviewer for this insightful comment. To address this comment, we examined the effects of the L1CAM antibody on tumor response and the expression L1CAM of tumour and normal cells, the results of which are shown in Supplementary Fig. 9 and 10 and discussed as below.

Results (p.17)

The anti-L1CAM antibody enhances radiation- and Dox-induced anti-tumour effects

To further validate the effects of the anti-L1CAM antibody on tumours, we investigated the effects of Ab417 treatment and radiotherapy or Dox treatment on tumour growth in nude mice bearing metastatic breast carcinoma MDA-MB-231 xenografts. Compared to that in non-treated tumours, L1CAM expression significantly increased in MDA-MB-231 tumours treated with either irradiation or Dox (Supplementary Fig. 9a and b). Compared to IgG control mice, Ab417-treated mice exhibited approximately 41% tumour growth inhibition at 30 days after irradiation (Supplementary Fig. 9a, bottom). Additionally, Ab417 treatment significantly enhanced Dox-induced anti-tumour effects (Supplementary Fig. 9b, bottom). These results indicate that Ab417 increases anti-tumour effects in combination with radiation therapy and Dox treatment.

Additionally, to address whether the effects of L1CAM on cancer cells differed from those on normal ECs, we compared L1CAM expression in MDA-MB-231 cells and HUVECs. L1CAM expression was found to be higher in MDA-MB-231 cells than in HUVECs (Supplementary Fig. 10a and b). However, EC-L1CAM was highly suppressed under normal conditions and upregulated later (48 h after irradiation) (Supplementary Fig. 10a). In contrast to MDA-MB-231 cells, in HUVECs, we found that Myc expression was peaked at 1hr after irradiation during DNA damage checkpoint signalling, and then degraded. At 48 h, L1CAM expression was significantly induced, and Myc expression was again increased in normal ECs (Supplementary Fig. 10b). These different L1CAM expression may contribute to the contrasting roles of L1CAM in cancer cells and normal ECs. From these results, we suggest that Ab417 not only can prevent DNA damage-induced cardiotoxicity without interfering with the anti-tumour effects, rather than enhancing anti-tumour effects.

The following paragraph was added to the Discussion section (p. 21):

Our data also showed that Ab417 markedly enhanced the anti-tumour effects of radiation and Dox treatment, indicating that Ab417 can prevent cardiotoxicity by enhancing the efficacy of anticancer therapy. However, the differential function of L1CAM in tumor and normal cells remains undefined. It seems that cancer cells including MDA-MB-231 cells exhibit aberrant expression of L1CAM. Furthermore, in glioblastoma multiforme (GBM) cancer cells, highly expressed L1CAM was previously found to increase DNA damage checkpoint activation by NBS1 upregulation via c-MYC 3 h after radiation, resulting in radioresistance³³. However, in ECs, L1CAM and c-Myc expression was upregulated after induction, and L1CAM and r-H2AX colocalized in the late stage (48 h after radiation) in our study, indicating persistent DNA damage. According to recent reports, Myc influences genomic instability in normal cells^{67, 68}. Therefore, we cautiously hypothesize that the opposite responses of L1CAM in tumour cells and normal ECs may be due to the dual nature of Myc (i.e., checkpoint signalling activation and survival in cancer cells vs genomic instability in normal cells). However, the nature of these differential mechanisms of L1CAM and the precise interactions between L1-CT and colocalized DNA damage repair proteins remain unclear, and we are currently investigating these topics in another study.

Moreover, early markers of activation of DNA damage response are decreased in both the endothelial cell (this paper) and cancer cell (ref 30) contexts.

Loss of L1CAM signaling in tumor cells inhibited early DNA damage checkpoint activation, showing decreased early markers of DNA damage response to overcome radioresistance. However, in normal ECs, although 53BP1, DNA-PKcs and p-ATM were recruited to persistent DNA damage sites, DNA damage checkpoint was not efficiently activated. We cautiously hypothesize that the loss of L1CAM decreased persistent DNA damage, not early DNA damage response, and subsequently exhibited decreased recruitment of 53BP1, DNA-PKcs and p-ATM to persistent DNA damage foci.

3. The next hole in mechanism concerns how endothelial cell DNA damage brings about cardiomyocyte dysfunction. To the authors' credit, they undertook co-culture experiments with stressed endothelial cells and human iPS cell-derived cardiomyocytes. The authors posit that endothelial DNA damage impacts cardiomyocytes through changes in extracellular matrix and hypoxia. But the data presented are insufficient in my mind to define any mechanism. There are also some related concerns. First, iPS cell-derived cardiomyocytes are very immature cells and may not be the correct cells with which to be asking this question in. Second, the loss of troponin from the cardiomyocytes, which the authors interpret as cardiomyocytes dysfunction, may well reflect cardiomyocyte death. Third, the data supporting that the effects of radiation are limited to endothelial cells – not cardiomyocytes – are not so clear.

Response: Per the reviewer's insightful inquiry, we have added the data as follows;

- First, to investigate the immature characteristics of iPS cells, additional experiments were performed in human primary adult cardiomyocytes, as shown in Supplementary Fig. 8e.
- To address your second point, we examined whether the loss of troponin in cardiomyocytes was caused by cardiomyocyte death using TUNEL staining (Supplementary Fig. 8c). Apoptotic cells were not detected in cardiomyocytes but were detected in 7.5 % of irradiated ECs (Supplementary Fig. 8c).
- Third, we attempted to define the effects of ECs on irradiated cardiomyocytes, irradiated human adult cardiomyocytes were cultured with conditioned medium from non-irradiated and irradiated ECs (Supplementary Fig. 8e).

The text in the Results section (p. 16) was added as follow;

To better define how irradiated ECs affect cardiomyocyte dysfunction, 10Gy-irradiated human adult cardiomyocytes were cultured with conditioned medium from non-irradiated and 10Gy-irradiated ECs (Supplementary Fig. 8e). Conditioned medium from non-irradiated ECs significantly increased the expression levels of brain natriuretic peptide (BNP) and cardiac troponin I (TNI) in irradiated human adult cardiomyocytes, but conditioned media from irradiated ECs did not (Supplementary Fig. 8e),

suggesting that secretory factors from ECs may affect irradiated cardiomyocyte dysfunction following irradiation. It has been reported that Neuregulin-1 (NRG-1), a angiocrine secreted from cardiac EC, plays a critical role in the protection of cardiomyocytes^{52, 53}. NRG-1 increased BNP and TNI expression in irradiated cardiomyocytes (Supplementary Fig. 8e). Coincidentally, endothelial-specific NRG-1 expression in irradiated hearts was prominently decreased compared to that in non-irradiated hearts, whereas Ab417-treated irradiated hearts did not exhibit decreased NRG expression (Supplementary Fig. 8f). Furthermore, RNA-seq analysis of Fig. 1c showed that the expression of NRG-1 was significantly reduced in 10 Gy-irradiated cells 72 h after irradiation compared to 10 Gy-irradiated cells 5 h after irradiation or in non-irradiated cells (Supplementary Fig. 8g). From these results, we suggest that irradiated ECs reduce cardiac angiocrine NRG-1 to protect against irradiated cardiomyocyte dysfunction, whereas Ab417 treatment can restore this reduced angiocrine function.

We have added the text in the Discussion section (p. 24) as follows;

Additionally, we focused on the effects of cardiac angiocrine factors, such as NRG-1, on cardiomyocyte protection. It has been reported that cardiac angiocrines, secretory factors of ECs, mediate EC-CM interactions by paracrine signaling⁶³. Of these cardiac angiocrines, NRG-1 is known to protect cardiomyocytes from ischaemic injury and chemotherapeutic agents^{52, 53}. Secretory factors from normal ECs, but not those from irradiated ECs undergoing EndMT, can reduce cardiomyocyte damage. In particular, our findings suggest that the angiocrine NRG-1 can protect cardiomyocytes from irradiation.

4) Specific comments on the data:

a. Fig.1c label on graph says 72h whereas main text says 48h.

Response: The text in the Results section (p.8) was revised to 72 h.

b. Fig.1d if L1CAM colocalization with γ -H2AX foci orchestrates persistent DNA damage, why DNA DSB goes down in 24h post IR while number of L1CAM foci goes up?

Response: We apologize that the original text was confusing regarding quantification of γ -H2AX foci number. We have addressed this by providing clearer explanations as below and additional data showing the criteria used to count the number of foci in Supplemental Fig 2a. Accordingly, we have corrected the graph of γ -H2AX foci number (Fig. 1d). Additionally, L1CAM expression was highly increased 48 hr after radiation and nuclear foci were dominantly colocalized with γ -H2AX foci at 48 hr, not 24 hr to regulate persistent DNA damage.

The text in the Results section (p. 7) was added as follow;

The average number of γ -H2AX foci per cell at least 70 cells per field was increased from 1 to 59 at 1 h post-irradiation and decreased to 25 at 48 h post-irradiation, indicating persistent DNA damage in the cells (Fig. 1d), whereas the intensity and foci diameter were significantly decreased at 24 h and after increased again thereafter (Supplemental Fig. 2a). The number of γ -H2AX foci with an intensity larger than 40 and a foci diameter of 0.1 μ m was counted.

We have added the text in the Discussion section (p. 19) as follows;

In this study, we showed that the intensity and size of foci were increased after 24 h of irradiation. This finding might reflect the minor population of DNA damage from the irradiation as well as the outcome of secondary cellular processes. The inability to repair DNA damage at telomeric regions is a major contributor to a persistent DDR⁵⁴. Moreover, DSB can also be formed indirectly from base excision repair (BER) of clustered lesions, DNA replication of SSB-containing templates or from transcription processes⁵⁵. Therefore, various causes might contribute to the late occurrence of γ -H2AX foci.

c. Fig.1f is the quantification done on the cytosolic, nuclear or total level?

Response: We have added a more detailed explanation to the legend of Figure 1f.

Quantification of full-length L1CAM was performed in the cytoplasmic fractions and L1-CT fragments in the nuclear fractions.

d. Phalloidin stains F-actin. Why are you using it as a marker of fibrosis?

Response: Fibroblastic cells exhibit an elongated mesenchymal cell phenotype accompanied by actin cytoskeleton reorganization. Thus, radiation-induced EndMT exhibited a fibrotic phenotype with phalloidin staining.

The text in the Results section (p. 7) was revised as follow; The severity of the elongated fibrotic phenotype (marked by phalloidin, showing actin cytoskeleton reorganization) in human umbilical vein endothelial cells (HUVECs) was correlated with the number of DNA damage foci (Fig. 1a).

e. Fig.1g and 1j, should also plot individually number of γ -H2AX foci and number of L1CAM foci.

Response: We have added the individual numbers of γ -H2AX foci and L1CAM foci to Fig. 1g and 1i in the revised figures

f. Supp Fig.1d surprising how Ab417 is able to limit DNA damage but has no effect on cell death. Should test more time points and also test in doxorubicin model.

Response: We performed FACs analysis in Supplementary Fig. 1 as follow; In fact, 72 h or 120 h after irradiation, Ab417 did not significantly affect cell death but reduced the number of aneuploid cells (Supplementary Fig. 2d). Ab417 also did not significantly affect Dox-induced cell death (Supplementary Fig. 2h).

In addition, the results in the Supplementary Fig. 10A showed that L1CAM in ECs is upregulated in the late phase (48 h after irradiation), not in the early phase of DNA damage response-related cell death. Moreover, L1CAM in ECs is suggested to sustain the persistent genomic instability.

g. Supp Fig.1e legend duplication error.

Response: We thank the reviewer for pointing this out; we have corrected this error.

h. Fig. 5. FS (one dimension) and EF (3 dimensions) are essentially the same parameter. Need to also specify LVESV and LVEDV.

Response: We have added quantitative data on LVESV (μl) and LVEDV (μl) to Fig. 5d and g.

Results (p.16) ; Echocardiography showed that Ab417 markedly prevented radiation-induced LV dysfunction, as the FS and left ventricular ejection fraction (LVEF) were higher in Ab417-treated mice than in IgG-treated mice. However, the left ventricular end systolic volume (LVESV) and left ventricular end diastolic volume (LVEDV) were lower in Ab417-treated mice than in IgG-treated mice (Fig. 5d).

5) Not completely accurate that no therapeutic targets have been identified in doxorubicin-associated cardiomyopathy. See Sci Transl Med, 2019. 11 (478): eaau8866. PMID: c. And Nat Cancer, 2020. 1(3): 315-328.PMID:32776015.

Response: In agreement with the reviewer's comment, the text in the Discussion section, "However, therapeutic targets to control or attenuate this damage have not yet been identified", was modified as follows and moved to the Introduction section of revised manuscript (p.5).

Introduction (p.5) ; Targeting the molecular pathways underlying this cardiotoxicity could help prevent or control such cardiac complications after anticancer therapy. As a chelator of intracellular iron, dexrazoxane, an FDA-approved drug, prevents Dox-induced heart failure from interfering with the anti-tumour effect of Dox⁸. A recent study also reported that a small molecule allosteric inhibitor of BAX and a compound that stabilizes tetrameric PKM2 prevent Dox-induced cardiomyopathy^{9, 10} ; However, there are still no clinical therapies to efficiently prevent Dox-induced cardiotoxicity

Reviewer #2 (Remarks to the Author):

The authors applied the anti-L1CAM antibody (Ab417, affinity matured Ab4) to attenuate cardiotoxicity after treatment by thoracic irradiation and doxorubicin. They found that vascular endothelial cells showed fibrotic phenotype cells after treatment and that L1-CAM was colocalized with DNA damage foci and it was involved in persistent DNA damage. Their strategy is that administering an anti L1CAM antibody can reduce the overall expression level of L1-CAM on the surface of cardiovascular or vascular ECs and it may lead to a decrease in the number of DNA damage foci.

Comments

1) The authors suggested that Ab417 may inhibit endocytosis of L1CAM, but they should clarify where the Ab418-L1CAM complex is. While the complex remains on the surface of ECs, IgG1 isotype of Ab418 can lead to effector functions like ADCC, CDC, and ADCP, resulting in severe damage to ECs by immune cells and disrupting vasculature. Perhaps, authors need to conduct a mouse study with IgG4 isotype of Ab418 or LALA-PG mutants of IgG1 Fc to compare. In contrast, if the complex was not removed, it will be recycled by FcRn because the endothelial cells normally express FcRn to keep long-serum half-life of antibodies. So, the L1CAM expression level can be restored quickly unless Ab418 binding is pH-dependent. Need to address the possibility for those unwanted effects mediated by the anti-L1CAM antibody.

We apologize for the confusing representation of the results. To clarify the location of the Ab417-L1CAM complex, the endocytic internalization of Ab417 and lysosomal degradation of L1CAM were examined using pHrodo fluorescence dye and the lysosomal degradation inhibitor bafilomycin-A1. Additionally, to address the possibility of unwanted effects mediated by the anti-L1CAM antibody on endothelial cells, we performed ADCC assay in HUVECs as described below.

The text in the Results section (p. 11) was added as follows:

Ab417 promotes the endocytic internalization and lysosomal degradation of L1CAM

In accordance with our previous work³⁶, Ab417 inhibited the expression levels of full-length L1CAM and its C-terminal fragment in this study. L1CAM has also been reported to be regulated by its lysosomal degradation and endocytosis⁴⁹. To clarify how Ab417 directly affects L1CAM levels, we examined the effects of Ab417 on the endocytosis and lysosomal degradation of L1CAM. Ab417 was labelled with pHrodo fluorescence dye, which increases fluorescence upon internalization into acidic endosomes and lysosomes. The red fluorescence of internalized pHrodo-labelled Ab417 was observed at 1 h and peaked at 18 h after incubation in HUVECs (Supplementary Fig. 4a). Additionally, we examined whether Ab417 can directly induce the endocytotic internalization of endogenous L1CAM, resulting in lysosomal degradation. An anti-L1CAM antibody against endogenous L1CAM was labelled with green pHrodo dye. The pHrodo Green-labelled anti-L1CAM antibody colocalized with endocytic vesicles containing Ab417 with red fluorescence (Supplementary Fig. 4a), suggesting that

Ab417 induced the internalization and endocytosis of membrane L1CAM. Ab417 significantly reduced L1CAM expression in WT-L1CAM-overexpressing HUVECs, and this effect was inhibited by treatment with the lysosomal degradation inhibitor bafilomycin-A1 (Supplementary Fig. 4b and c). These results suggest that Ab417 promotes the internalization, endocytosis and subsequent lysosomal degradation of L1CAM in HUVECs.

Additionally, to address the possibility of unwanted effects by Ab417, we performed an NK cell-mediated antibody-dependent cellular cytotoxicity (ADCC) assay to determine whether Ab417 on HUVECs can lead to effector function. Ab417 on HUVECs did not trigger the ADCC effect, and the ADCC effect of the anti-CD20 antibody on Raji cells was used as a positive control (Supplementary Fig. 4d). This result supports our data that Ab417 protects against EC damage following radiation and Dox treatment without ADCC effects.

‘Together, our findings suggest that Ab417 reduced L1CAM expression by the endocytotic lysosomal degradation of full-length L1CAM and subsequently reduced the expression of the L1-CT fragment. Therefore, the Ab417 antibody inhibits persistent DNA damage by reducing the expression of full-length L1CAM and the subsequent nuclear L-CT fragment and does not directly interfere with the cleavage and nuclear translocation of L1CAM.’

Thus, our incorrect interpretation of Supplementary Fig 2a of original text “Ab417 may inhibit endocytosis of L1CAM” was deleted. From the results of the new experiments, we now suggest that the reduced colocalization of LAMP1 and L1CAM is due to L1CAM lysosomal degradation by Ab417 treatment.

2) I would suggest discussing which mechanism enhanced L1CAM expression after irradiation and Dox treatment. The increased expression level of L1CAM can be also part of natural mechanisms to prevent cell growth and induce cell death as DNA-damaged cells can transform to tumor cells. In this context, did the EC or other cells treated by the anti-L1CAM antibody have normal DNA after repairing or still have damaged DNA? I am concerned that L1CAM antibody treatment can lead to tumor generation.

Response: Per the reviewer’s insight inquiring, we have now examined DNA damage and aneuploidy cell populations in prolonged culture of Ab417-treated normal cells after irradiation to investigate the effects of anti-L1CAM antibody on tumor generation (Supplementary Fig 2f).

The text in the Results section (p. 8) was added as follows:

Additionally, we found that in prolonged culture (up to 40 days following irradiation), Ab417-treated ECs prominently sustained reduced DNA damage and aneuploidy compared to IgG-treated ECs. The number and intensity of L1-CT foci were also decreased at longer time points in Ab417-treated cells, compared to IgG-treated ECs (Supplementary Fig 2f).

Discussion (p.20): Ab417 regulates the L1CAM expression level which regulates malignant transition. L1CAM expression is tightly regulated in chemoresistant cells undergoing EMT⁵⁴. HIF-1a and TGF- β 1 are EMT regulators that also regulate L1CAM expression in cancer cells⁵⁵. TGF- β 1-induced L1CAM expression is related to apoptotic resistance and promotes the malignant transition of intestinal epithelial cells⁵⁶. Our previous study showed that the radiation-induced EndMT process is mainly regulated by HIF-1a and TGF- β 1 in normal ECs³⁵. Thus, increased L1CAM expression after radiation may also be a part of natural mechanisms to survive with irreparable DNA damage, promoting EndMT. Ab417-treated cells exhibited reduced radiation-induced aneuploidy of normal cells, suggesting that Ab417 can regulate tumorigenic changes in DNA-damaged normal cells.

3) The author may need to demonstrate actual benefits in tumor-bearing mouse models, not in normal mice. I wonder whether the Dox and irradiation enhance the expression level of L1CAM on tumor cells because the L1CAM antibody treatment also can give a benefit to tumor cells for surviving through the same mechanism as for the ECs.

Response: Thank you for your insightful comment. To address this comment, we examined the effects of the L1CAM antibody on tumor response and the expression L1CAM of tumour and normal cells, the results of which are shown in Supplementary Fig. 9 and 10 and discussed as below.

Results (p.17)

The anti-L1CAM antibody enhances radiation- and Dox-induced anti-tumour effects

To further validate the effects of the anti-L1CAM antibody on tumours, we investigated the effects of Ab417 treatment and radiotherapy or Dox treatment on tumour growth in nude mice bearing metastatic breast carcinoma MDA-MB-231 xenografts. Compared to that in non-treated tumours, L1CAM expression significantly increased in MDA-MB-231 tumours treated with either irradiation or Dox (Supplementary Fig. 9a and b). Compared to IgG control mice, Ab417-treated mice exhibited approximately 52% tumour growth inhibition at 30 days after irradiation (Supplementary Fig. 9a). Additionally, Ab417 treatment significantly enhanced Dox-induced anti-tumour effects (Supplementary Fig. 9b). These results indicate that Ab417 increases anti-tumour effects in combination with radiation therapy and Dox treatment.

Additionally, to address whether the effects of L1CAM on cancer cells differed from those on normal ECs, we compared L1CAM expression in MDA-MB-231 cells and HUVECs. L1CAM expression was found to be higher in MDA-MB-231 cells than in HUVECs (Supplementary Fig. 10a and b). However, EC-L1CAM was highly suppressed under normal conditions and upregulated later (48 h after irradiation) (Supplementary Fig. 10a). In contrast to MDA-MB-231 cells, in HUVECs, we found that Myc expression was peaked at 1hr after irradiation during DNA damage checkpoint signalling, and then degraded. At 48 h, L1cam expression was significantly induced, and Myc expression was again increased in normal ECs (Supplementary Fig. 10b). These different L1CAM expression may contribute to the contrasting roles of L1CAM in cancer cells and normal ECs. From these results, we

suggest that Ab417 not only can prevent DNA damage-induced cardiotoxicity without interfering with the anti-tumour effects, rather than enhancing anti-tumour effects.

The following paragraph was added to the Discussion section (p. 22): “Our data also show that Ab417 markedly enhanced the anti-tumour effects of radiation and Dox treatment, indicating that Ab417 can prevent cardiotoxicity by enhancing the efficacy of anticancer therapy. However, the differential function of L1CAM in tumor and normal cells remains undefined. It seems that cancer cells including MDA-MB-231 cells exhibit aberrant expression of L1CAM. Furthermore, in glioblastoma multiforme (GBM) cancer cells, highly expressed L1CAM was previously found to increase DNA damage checkpoint activation by NBS1 upregulation via c-MYC 3 h after radiation, resulting in radioresistance³³. However, in ECs, L1CAM and c-Myc expression was upregulated after induction, and L1CAM and r-H2AX colocalized in the late stage (48 h after radiation) in our study, indicating persistent DNA damage. According to recent reports, Myc influences genomic instability in normal cells^{67, 68}. Therefore, we cautiously hypothesize that the opposite responses of L1CAM in tumour cells and normal ECs may be due to the dual nature of Myc (i.e., checkpoint signalling activation and survival in cancer cells vs genomic instability in normal cells). However, the nature of these differential mechanisms of L1CAM and the precise interactions between L1-CT and colocalized DNA damage repair proteins remain unclear, and we are currently investigating these topics in another study.”

Reviewer #3 (Remarks to the Author):

The paper by Nam and colleagues show that vascular endothelial cells after irradiation or doxorubicin show persistent DNA damage foci, co-localizing with C-terminal domain of LICAM. Additionally, they show that treatment with anti-LICAM antibody decreased LICAM overexpression and nuclear translocation in VECs, as well as reduced the number of DNA damage foci. After whole-heart irradiation, VEC-specific p53 deletion increased LICAM/DNA damage foci colocalization and vascular fibrosis, but these effects were attenuated by Ab417 treatment- this also led to improvements in cardiac function and prolonged survival. Overall, It is a potentially important study (which provides both some mechanistic insight as well as therapeutic opportunity) and the authors have used multiple approaches to support their conclusions (using both in vitro and in vivo models). Mechanistically, I think that there are several gaps which still need to be addressed- particularly the relationship between DDR and vascular fibrosis and the detailed mechanisms by which LICAM impacts on DNA repair. However, I think no single paper can address all questions. Another point is the apparent proposal that effects of irradiation are specific to vascular cells in the heart (it would be difficult to understand why other cell-types in the heart would not be able to mount a DDR after such as high dose of irradiation).

There are some aspects which I believe need to be addressed so that publication can be supported.

Figure 1- The authors do not show convincing evidence that the DDR foci are persistent. All data relies on quantification of DDR foci in fixed cells at different time points. In order to make a point about a persistent DNA damage response – DDR reporters and live-cell imaging needs to be performed allowing the tracking of the lifespan of individual foci. Additionally, longer time points should be analyzed- see for instance previous reports have indicated that DDR foci can be persistent even months following irradiation (PMID: 22426077; PMID: 22426229). The kinetic window provided is too narrow.

Response: Thank you for your insightful comments. To examine the radiation-induced DNA damage response, we traced DNA damage foci in HUVECs expressing GFP-53BP1 after radiation using live cell imaging (Supplementary Fig. 2b). Additionally, due to technical limitations of photobleaching and phototoxicity in single-spot scanning confocal microscopy, we could not trace individual foci beyond 48 h after radiation; therefore, we analysed the survival curve of >300 foci from 10 cells up to 48h after radiation. Additionally, we found that DDR foci can persist for up to 40 days following irradiation using fixed cells (Supplementary Fig. 2f).

The text in the Results section (p. 7, 8) was added as follows: Live cell imaging was used to confirm the persistence of DDR foci, and 53BP1 DNA damage foci were traced in HUVECs transfected with GFP-53BP1 (a marker of DNA damage foci) ; Kaplan-Meier survival curves for GFP-positive foci were analysed from 12 h after 10 Gy irradiation, showing that ~% of GFP-positive foci had a lifespan of over (Supplementary Fig 2b).

Additionally, we found that in prolonged culture (up to 40 days following irradiation), Ab417-treated ECs prominently sustained reduced DNA damage and aneuploidy compared to IgG-treated ECs. The number and intensity of L1-CT foci were also decreased at longer time points in Ab417-treated cells, compared to IgG-treated ECs (Supplementary Fig 2f).

Also, I am wondering how the authors explain that the DDR decreased at 24h and then increased again at 48h after 10Gy irradiation (this would, in my view, indicate that these are newly generated DDR foci – possibly the outcome of secondary processes and not the ones generated as a result of the radiation).

Response: To address the reviewer's comment, we have revised the text in the Results section (p. 7) as follows. We have provided additional data in Supplemental Fig 2a, showing the criteria used to count the number of foci.

The text in the Results section (p. 7) was added as follows:

The average number of γ -H2AX foci per cell at least 70 cells per field was increased from 1 to 59 at 1 h post-irradiation and decreased to 25 at 48 h post-irradiation, indicating persistent DNA damage in the cells (Fig. 1d), whereas the intensity and foci diameter were significantly decreased at 24 h and after increased again thereafter (Supplemental Fig. 2a). The number of γ -H2AX foci with an intensity larger than 40 and a foci diameter of 0.1 μ m was counted (Fig. 1d).

We have added the text in the Discussion section (p. 19) as follows:

In this study, we showed that the intensity and size of foci were increased after 24 h of irradiation. This finding might reflect the minor population of irreparable DNA damage from the irradiation as well as the outcome of secondary cellular processes. The inability to repair DNA damage at telomeric regions is a major contributor to a persistent DDR⁵⁴. Moreover, DSB can also be formed indirectly from base excision repair (BER) of clustered lesions, DNA replication of SSB-containing templates or from transcription processes⁵⁵. Therefore, various causes might contribute to the late occurrence of γ -H2AX foci.

- KD efficiency of L1CAM by siRNA is very mild (western blot not particularly convincing in my view). Have the authors considered other potentially more effective (non transient) ways to KD L1CAM (for instance using lentiviral mediated shRNA or CRISPR/CAS9). I am also not entirely sure what was the point of showing the siRNA experiments- if no other outcomes were shown in the main Figure.

Response: We thank the reviewer for this apt comment and we replaced the data regarding the lentiviral shRNA of L1CAM in Fig 1e. Additionally, we added data on L1-CT and γ -H2AX foci colocalization to Fig 1e. We used the lentiviral shRNA to knockdown L1CAM to show that the L1-CT and γ -H2AX colocalization disappeared, indicating that this colocalization was dependent on L1CAM expression.

(minor points)

- In the text authors indicate that RNA-seq showed increased expression of EndMT-related genes at 48h- however, in the Figure 1c 72h is indicated.

Response : The text in the Results section (p.7) was revised to 72 h.

- Co-localization of L1CAM and gH2A.X is not evident in small Figure panels- I would advise showing amplified co-localized foci and indicating co-localization with arrows.

Response: We thank the reviewer for pointing this out. We have added the amplified colocalized foci of L1CAM and γ -H2AX to Fig. 1 and 2.

- Quality of Immunofluorescence in dox treatment is not particularly good- no clear large L1CAM foci are observed- why is that?

Response: In agreement with the reviewer's comment, we replaced another image with clear L1CAM foci (Fig. 1i).

However, the sizes of Dox-induced DNA damage foci were irregular compared to those of radiation-induced DNA damage foci, and these differences may be due to radiation directly inducing DNA breaks, whereas Dox interacts with DNA through intercalation (Fig. 1g and i).

- Could other confirmatory methods such as Immunoprecipitation be performed to further confirm the association between gH2A.X and L1CAM?

Response: To address the reviewer's comment, an immunoprecipitation assay was performed, as shown in Supplementary Fig. 3e. This experiment confirmed that L1-CT binds to r-H2AX in irradiated ECs transfected with GFP-L1-CT.

- Effects of Ab417 on phalloidin expression are relatively mild- are there effects of Ab417 cell on size (it seems so from pictures)? What about cell proliferation- since DDR is affected it is reasonable to assume that cells may overcome cell-cycle arrest induced by DNA damage.

Response: Thank you for your insightful comment. We added a more detailed explanation of the effects of Ab417 on cell size as described in the following passage from the Results (p.8) section: "Furthermore, Ab417 pre-treatment resulted in a significant reduction in the elongated fibrotic phenotype of HUVECs 72 h after irradiation, showing a reduced size of irradiated ECs with flattened nuclei compared to non-irradiated ECs (Fig. 1h and Supplementry Fig. 2C)."

Additionally, we examined the effects of Ab417 on the proliferation of irradiated ECs using an EdU assay as shown in the following Results (p.8) subsection: “ The proliferation rate of Ab417-treated ECs was significantly higher than that of IgG-treated ECs at 72 and 120 h after irradiation (Supplementary Fig. 2d).”

Figure 2- b) Western showing LICMA expression following transfection- loading control is necessary. Also, the necessary controls should be added? Authors mentioned they have used lentivirus- mediated shRNA against LICAM and then reintroduced LICAM WT or mutant forms- however, there is no evidence of KD efficiency. Cells expressing a shRNA control should be included in the same western blot.

Response: We thank the reviewer for pointing this out. We have added the loading control “ β -actin” and the immunoblot containing the shRNA control to Fig. 2b.

Immunofluorescence images for pATM and 53BP1- authors should better indicate co-localize foci (possibly showing magnifications). Why is p-ATM cytosolic in no IR cells (Supplementary Figure 2)? Quality of DNA-PKcs immunofluorescence is not ideal (I am not sure how the authors were capable of detecting/quantifying individual foci). Other methods for determining co-localization should be employed here (Pearson’s correlation coefficient?)

Response: In Figure 2d, we marked the colocalized foci with arrows and added the amplified colocalized foci of p-ATM, 53BP1, DNA-PKcs and L1-CT. For the exact quantification of colocalized foci of DNA-PKcs with L-CT, we replaced the graph of the colocalized foci number with Pearson’s correlation coefficient.

What do the authors mean with (Line247) persistent DNA damage repair?

Response: We have corrected the text of original manuscript (Line247: The anti-L1CAM antibody enhances persistent DNA damage repair.) in the Results (p.12) section as follows : The Ab417 antibody inhibits persistent DNA damage by reducing the expression of full-length L1CAM.

Typo (line 197) We replaced ‘for and L1-CT and r-H2AX’ with ‘for L1-CT and r-H2AX’.

Figure 3- Authors show that knock-down of p53 in HUVECs show increased L1CAM expression and increased co-localization with gH2A.X after IR. I am wondering if p53KD impacts on cell death and proliferation following IR. (this point is also relevant for the in vivo experiments in Figure 3 and 4)

Response: Results (p.12): According to FACS analysis, in HUVECs, p53 knockdown increased radiation-induced cell death and decreased cell proliferation reduced by irradiation, and these effects were reversed in Ab417-treated cells (Supplementary Fig 5b).

d- immunofluorescent images for L1CAM by IF in hearts from wt and p53KO mice are not very convincing. Is there increased L1CAM/gH2A.X co-localization?

Response: We thank the reviewer for pointing this out. We have changed the representative images for L1CAM in Fig.3d. In addition, we added the images and quantification graphs of the colocalization of L1-CT and r-H2AX foci in Supplementary Figure 5d.

Results (p.13): At 3 weeks after whole-heart irradiation, DNA damage, nuclear L1-CT and the colocalization of L1-CT and γ -H2AX in cardiovascular ECs of control IgG-treated EC-p53KO mice were increased compared to those of wild-type mice. Importantly, these effects in EC-p53KO irradiated mice were reversed by Ab417 treatment (Fig. 3d and Supplementary Fig. 5d and e).

Supplementary Figure 3- Authors claim that DDR is not present in cardiomyocytes. Quality of the gH2A.X and WGA staining shown is not particularly convincing. Besides this is at odds with other reports. Activation of a DDR in cardiomyocytes has been shown to be a mechanism contributing to their post-mitotic state during differentiation PMID: 24766806. gH2A.X foci did not change with age and were detected in almost all adult CMs irrespective of age PMID: 30737259. Thus, even if gH2A.X was not induced in cardiomyocytes following irradiation (which is surprising and not sure why it would happen)- it should be present in adult cardiomyocytes irrespectively. I wonder if this is a technical issue.

Response: We apologize for the confusing representation of the results. To better clarify our finding, we have changed the representative images including enlarged images and added a graph showing the relative intensity of r-H2AX foci in ECs and CMs. In the hearts of WT and p53KO mice, radiation-induced DNA damage was observed in both ECs and CMs; however, DNA damage in vascular ECs was more prominent than that in CMs as we describe in the following Results:

Results (p.13) : “ In irradiated cardiac tissues of WT mice, ECs (7.1-fold) were more γ -H2AX-positive than cardiomyocytes (3.4-fold), compared to non-irradiated cardiac tissues, and these phenotypes were enhanced in the irradiated cardiac tissues of pEC-53 KO mice. Additionally, Ab417 reduced DNA damage in ECs to a greater extent than CMs in the irradiated cardiac tissues of pEC-53 KO mice (Supplementary Fig. 5f).”

Figure 6- can statistics be performed with n=2 controls?

Response: To improve the accuracy of the quantification, we analysed more 6 normal cardiac issues from healthy patients, recalculated the relative intensities and added one more representative image of healthy tissue to Fig 6.

Reviewers' Comments:

Reviewer #1:

Remarks to the Author:

No further comments.

Reviewer #2:

Remarks to the Author:

The authors addressed my concerns

Reviewer #3:

Remarks to the Author:

I thank the authors for addressing my comments and including several new pieces of data. I think the manuscript has improved considerably. The data with regards to the persistence in DDR foci is more convincing as well as some of the downstream effects.

I still believe that quality of the immunostaining in tissue sections is not very convincing, as it is difficult to visualize gH2A.X or L1CAM foci.

Functional measurements of cardiac function and survival curves make a compelling case for the therapeutic potential of Ab417 treatment.

Responses to the Reviewer's comments

Reviewer #3 (Remarks to the Author):

I thank the authors for addressing my comments and including several new pieces of data. I think the manuscript has improved considerably. The data with regards to the persistence in DDR foci is more convincing as well as some of the downstream effects.

I still believe that quality of the immunostaining in tissue sections is not very convincing, as it is difficult to visualize γ H2A.X or LICAM foci.

Functional measurements of cardiac function and survival curves make a compelling case for the therapeutic potential of Ab417 treatment.

Response: Thank you for your comment. We did recognize the concern. However, as being pointed out, it is challenging to clearly visualize γ -H2AX or LICAM foci in tissue sections (rather clear for in vitro cells) and we made a best case by using available tissue sections in this study. Although it is difficult to resolve this issue for now, methodological improvements to address it should be made in the further studies.